

# Source attribution using FLEXPART and carbon monoxide emission inventories: SOFT-IO version 1.0

Bastien Sauvage[1], Alain Fontaine[1], Sabine Eckhardt[3], Antoine Auby[4], Damien Boulanger[2], Hervé Petetin[1], Ronan Paugam[5], Gilles Athier[1], Jean-Marc Cousin[1], Sabine Darras[3], Philippe Nédélec[1], Andreas Stohl[3], Solène Turquety[6], Jean-Pierre Cammas[7] and Valérie Thouret[1].

[1]Laboratoire d'Aérologie, Université de Toulouse, CNRS, UPS, France
[2]Observatoire Midi-Pyrénées, Toulouse, France
[3]NILU - Norwegian Institute for Air Research, Kjeller, Norway
[4]CAP HPI, Leeds, United Kingdom
[5]King's College, London, United Kingdom
[6]Laboratoire de Météorologie Dynamique/IPSL, UPMC Univ. Paris 6, Paris, France
[7]Observatoire des Sciences de l'Univers de la Réunion (UMS 3365) et Laboratoire de l'Atmosphère et des Cyclones (UMR 8105), Université de la Réunion, Saint-Denis, La Réunion, France

*Correspondence to*: Bastien Sauvage (bastien.sauvage@aero.obs-mip.fr)

**Abstract.** Since 1994, the In-service Aircraft for a Global Observing System (IAGOS) program has produced in-situ measurements of the atmospheric composition during more than 51000 commercial flights. In order to help analyzing these observations and understanding the processes driving the observed concentration distribution and variability, we developed the SOFT-IO tool to quantify source/receptor links for all measured data. Based on the FLEXPART particle dispersion model (Stohl et al., 2005), SOFT-IO simulates the contributions of anthropogenic and biomass burning emissions from the ECCAD emission inventory database for all locations and times corresponding to the measured carbon monoxide mixing ratios along each IAGOS flight. Contributions are simulated from emissions occurring during the last 20 days before an observation, separating individual contributions from the different source regions. The main goal is to supply added-value products to the IAGOS database by evincing the geographical origin and emission sources driving the CO enhancements observed in the troposphere and lower stratosphere. This requires a good match between observed and modeled CO enhancements. Indeed, SOFT-IO detects more than 95% of the observed CO anomalies over most of the regions sampled by IAGOS in the troposphere. In the majority of cases, SOFT-IO simulates CO pollution plumes with biases lower than 10-15 ppbv. Differences between the model and observations are larger for very low or very high observed CO values. The added-value products will help in the understanding of the trace-gas distribution and seasonal variability. They are available in the IAGOS data base via http://www.iagos.org. The SOFT-IO tool could also be applied to similar data sets of CO observations (e.g. ground-based measurements, satellite observations). SOFT-IO could also be used for statistical validation as well as for inter-comparisons of emission inventories using large amounts of data.

## 1 Introduction

Tropospheric pollution is a global problem caused mainly by natural or human-triggered biomass burning, and anthropogenic emissions related to fossil fuel extraction and burning. Pollution plumes can be transported



quickly on a hemispheric scale (within at least 15 days) by large scale winds or, more slowly (Jacob, 1999),
between the two hemispheres (requiring more than 3 months). Global anthropogenic emissions are for some
species ($CO_2$) in constant increase (Boden et al., 2015). However, recent commitments of some countries to
reduce greenhouse gas emissions (e.g. over the U.S., U.S. EPA's Inventory of U.S. Greenhouse Gas Emissions
and Sinks, 1990-2013; http://www.epa.gov/climatechange/ghgemissions/usinventoryreport.html) seems to
induce a stalling in other global emissions ($NO_x$, $SO_2$ and Black Carbon, Stohl et al., 2015), except for some
regions (Brazil, Middle East India, China) where $NO_x$ emissions increase (Miyazaki, 2017). In order to better
understand large-scale pollution transport, large amounts of in situ and space-based data have been collected in
the last three decades, allowing a better understanding of pollution variability and its connection with
atmospheric transport patterns (e.g. Liu et al., 2013). These data-sets are also useful to quantify global pollution
evolution with respect to the emissions trends described above.

Despite the availability of large trace gas data sets, the data interpretation remains difficult for the following

reasons: (1) the sampling mode does not correspond to an a priori defined scientific strategy, as opposed to data
collected during field campaigns; (2) the statistical analysis of the data can be complicated by the large number
of different sources contributing to the measured pollution, and an automated analysis of the contributions from
these different sources is required if, for instance, regional trends in emissions are to be investigated; (3) the
sheer size of some of the data sets can make the analysis rather challenging. Among the long-term pollution
measurement programs, the IAGOS airborne program (http://www.iagos.org/, formerly known as the
Measurement of OZone by Airbus In-service airCraft -MOZAIC- program) is the only one delivering in-situ
measurement data from the free troposphere. IAGOS provides regular global measurements of ozone ($O_3$) - since
1994 -, carbon monoxide (CO) - since 2002 -, and nitrogen oxides ($NO_y$) – for the period 2001-2005 - obtained
during more than 51000 commercial aircraft flights up to now, with substantial extent of the instrumented
aircraft recently. The analysis of the IAGOS database is also complicated by the fact that primary pollutants (CO
and part of $NO_y$) are emitted by multiple sources, while secondary compounds ($O_3$) are produced by
photochemical transformations of these pollutants, often most efficiently when pollutants from different sources
mix.

A common approach to separate the different sources influencing trace gas observations is based on the

determination of the air mass origins through Lagrangian modeling. This approach allows linking the emission
sources to the trace gas observations (e.g. Nédélec et al., 2005; Sauvage et al., 2005, 2006; Tressol et al. 2008;
Gressent et al. 2014; Clark et al., 2015; Yamasoe et al., 2015). Lagrangian modeling of the dispersion of
particles allows accounting efficiently for processes such as large-scale transport, turbulence and convection.
When coupled with emission inventories Lagrangian modeling of passive tracers allows for instance to
understand ozone anomalies (Cooper et al., 2006; Wen et al., 2012), to quantify the importance of lightning NOx
emissions for tropospheric $NO_2$ columns measured from space (Beirle et al., 2006), to investigate the origins of
$O_3$ and CO over China (Ding et al., 2013), or to investigate the sources influencing the observed $CO_2$ over the
high northern latitudes (Vay et al., 2011).

To help analyzing a large data set such as the IAGOS observations, it is important to provide scientific users

a tool for characterizing air mass transport and emission sources. This study presents a methodology to
systematically establish a link between emissions sources (biomass burning and anthropogenic emissions) and
concentrations at the receptor locations. Since CO is a substance that is emitted by combustion sources (both



anthropogenic and biomass burning) and since CO has a lifetime of months in the troposphere (Logan et al.,
1981; Mauzerall et al., 1998), it is often used as a tracer for pollution transport (Staudt et al. 2001; Yashiro et al.,
2009; Barret et al., 2016). It is therefore convenient to follow past examples and use simulated CO source
contributions to gauge the influence of pollution sources on the measurements also with SOFT-IO. Our
methodology uses the FLEXPART Lagrangian particle dispersion model (Stohl et al., 2005) and emission
inventories from the ECCAD emission database (Granier et al., 2012) in order to quantify the influence of
emissions sources on the IAGOS CO measurements. The goal is to provide the scientific community with added
value products that will help them analyzing and interpreting the large number of IAGOS measurements. The
methodology has the benefit to be adaptable to multiple emission inventories without re-running FLEXPART
simulations. It is also easily adaptable to analyze other datasets of trace gas measurements such as from ground
based observations, sondes, aircraft campaigns or satellite observations.

The methodology will be described in the next section, and then evaluated at the example of case studies of
pollution plumes observed by IAGOS aircraft. Further evaluation is performed through statistical analysis.
Finally we discuss the limitations of the methodology by estimating its sensitivity to different input data sets
(emission inventories, meteorological analyses).
**2. In-situ observations database: MOZAIC and IAGOS programs**
The MOZAIC program (Marenco et al., 1998) was initiated in 1993 by European scientists, aircraft
manufacturers and airlines to better understand the natural variability of the chemical composition of the
atmosphere and how it is changing under the influence of human activity, with particular interest in the impact of
aircraft exhaust. Between August 1994 and November 2014, MOZAIC performed airborne in-situ measurements
of ozone, water vapor, carbon monoxide, and total nitrogen oxides. The measurements are geolocated (latitude,
longitude and pressure) and come along with meteorological observations (wind direction and speed,
temperature). Data acquisition is performed automatically during round-trip international flights (ascent, descent
and cruise phases) from Europe to America, Africa, Middle East, and Asia (Fig. 1).
Based on the technical expertise of MOZAIC, the IAGOS program (Petzold et al., 2015, and references therein)
has taken over and provides observations since July 2011. The IAGOS data set still includes ozone, water vapor,
carbon monoxide, meteorological observations, and measurements of cloud droplets (number and size) are also
performed. Depending on optional additional instrumentation, measurements of nitrogen oxides, total nitrogen
oxides or, in the near-future, greenhouse gases ($CO_2$ and $CH_4$), or aerosols, will also be made.
Since 1994, the IAGOS-MOZAIC observations have created a big data set that is stored in a single database
holding data from more than 51000 flights. The data set can be used by the entire scientific community, allowing
studies of chemical and physical processes in the atmosphere, or validation of global chemistry transport models
and satellite retrievals. Most of the measurements have been collected in the upper troposphere and lower
stratosphere, between 9 and 12 km altitude, with 500 flights/ aircraft/ year on up to 7 aircraft up to now.

The MOZAIC and IAGOS data (called "IAGOS" from here on) used in this study are in-situ observations of CO
only, which is being measured regularly on every aircraft since 2002 with more than 30000 flights, using a
modified infrared filter correlation monitor (Nédélec et al., 2003; Nédélec et al., 2015). The accuracy of the CO
measurements has been estimated at (30 s response time) ± 5 ppb, or ± 5%.






Several case studies of CO pollution plumes (Table 1) using IAGOS data have been published, where model
simulations allowed attribution of the measured CO enhancements to anthropogenic or biomass burning
emissions, either measured in the boundary layer or in the free troposphere, following regional or synoptic-scale
transport (e.g. Nédélec et al., 2005; Tressol et al., 2008; Cammas et al., 2009; Elguindi et al., 2010). These case
studies are used here to better define the requirements for our methodology (meteorological analyses and
emission inventory inputs). Some of them are detailed and re-analyzed in Sect. 4.
**3. Estimation of carbon monoxide source regions: methodology**
To establish systematic source-receptor relationships for IAGOS observations of CO, the Lagrangian dispersion
model FLEXPART (Stohl et al., 1998, 2005; Stohl and Thomson, 1999) is run over the entire database.
Lagrangian dispersion models usually represent the differential advection better than global Eulerian models
(which do not well resolve intercontinental pollution transport; Eastham et al., 2017), at a significantly lower
computational cost. In particular, small-scale structures in the atmospheric composition can often be
reconstructed from large-scale global meteorological data, which makes model results comparable to high-
resolution in situ observations (Pisso et al., 2010). In the past, many studies (Nédélec et al., 2005 ; Tressol et
al.,2008; Cammas et al., 2009;  Elguindi et al., 2010; Gressent et al., 2014) used FLEXPART to investigate
specific pollution events observed by the IAGOS aircraft. However, in these former case studies, the link
between sources and observations of pollution was guessed a priori. The transport model was then used to
validate the hypothesis. For example, in the Cammas et al. (2009) study, observations of high CO during summer
in the upper troposphere and lower stratosphere east of Canada were guessed to originate from biomass burning
over Canada as this region is often associated with pyro-convection whose intensity usually peaks in the
summer. This origin was confirmed by the model analysis. In general, the origin of the observed pollution cannot
be guessed a priori, especially when analyzing measurements from thousands of flights. Moreover, multiple
sources are most of the time involved when the observed pollution is the result of the mixing of polluted air
masses from different regions and source types.
CO is often used as a tracer to quantify the contributions of the different sources to the observed pollution
episodes. CO is emitted by both the combustion of fossil fuels and by biomass burning, and its photochemical
lifetime against OH attack is usually 1 to 2 months in the troposphere (Logan et al., 1981; Mauzerall et al.,
1998). Therefore it is possible to link elevated CO mixing ratios (with respect to its seasonally varying
hemispheric baseline) to pollution sources without simulating the atmospheric chemistry.
**3.1 Backward transport modeling**
Simulations were performed using the version 9 of FLEXPART, which is described in detail by Stohl et al.
(2005) (and references therein). The model was driven using wind fields from the European Centre for Medium-
Range Weather Forecast (ECMWF) 6-hourly operational analyses and 3-hour forecasts. The ECMWF data are
gridded with a 1° × 1° horizontal resolution, and with a number of vertical levels increasing from 60 in 2002 to
137 since 2013. The model was also tested using higher horizontal resolution (0.5°), and with ECMWF ERA-
Interim reanalysis, as their horizontal and vertical resolution and model physics are homogeneous during the
whole period of IAGOS CO measurements. However, operational analyses were used for our standard set-up, as



the transport model reproduced CO better when using these data for several case studies of pollution transport,
especially for plumes located in the UT. Indeed, operational analyses provide a better vertical resolution since
2006 (91 levels until 2013, then 137 levels against 60 levels for ERA-Interim) and thus a better representation of
the vertical wind shear, and the underlying meteorological model is also more modern than the one used for
producing ERA-Interim. Vertical resolution is obviously the most critical factor for modeling such CO plumes
with the best precision in terms of location and intensity (Eastham and Jacob, 2017).
Using higher horizontal resolution for met-fields analyses and forecasts (0.5° vs 1°) showed no influence on the
simulated carbon monoxide, despite larger computational time and storage needs. We assume further
improvement can be obtained using even higher horizontal resolution (0.1°), but this was not feasible at this
stage and should be considered in the future.

In order to be able to represent the small-scale structures created by the wind shear and observed in many
IAGOS vertical profiles, the model is initialized along IAGOS flight tracks every 10 hPa during ascents and
descents, and every 0.5° in latitude and longitude at cruise altitude. This procedure leads to $i$ model initialization
boxes along every flight track. For each $i$, 1000 particles are released. Indeed 1000 to 6000 particles are
suggested for correct simulations in similar studies based on sensitivity tests on particles number (Wen et al.,
2012; Ding et al., 2013). For instance, a Frankfurt (Germany) to Windhoek (Namibia) flight contains around 290
boxes (290000 particles) of initialization as a whole.
FLEXPART is set up for backward simulations (Seibert and Frank, 2004) from these boxes as described in Stohl
et al. (2003) and backward transport is computed for 20 days prior to the in-situ observation, which is sufficient
to consider hemispheric scale pollution transport in the mid-latitudes (Damoah et al., 2004; Stohl et al., 2002;
Cristofanelli et al., 2013). This duration is also expected to be longer than the usual lifetime of polluted plumes
in the free troposphere, i.e. the time when the concentration of pollutants in plumes is significantly larger than
the surrounding background. Indeed, the tropospheric mixing time scale has been estimated to be typically
shorter than 10 days (Good et al., 2003; Pisso et al., 2009). Therefore the model is expected to be able to link air
mass anomalies such as strong enhancements in CO to the source regions of emissions (Stohl et al., 2003). It is
important to note that we aim to simulate recent events of pollution explaining CO enhancements over the
background, but not to simulate the CO background which results from aged and well-mixed emissions.
The FLEXPART output is a residence time, as presented and discussed in Stohl et al. (2003). These data
represent the average time spent by the transported air masses in a grid cell, divided by the air density, and are
proportional to the sensitivity of the receptor mixing ratio to surface emissions. In our case, it is calculated for
every input point along the flight track, every day for $N_t = 20$ days backward in time, on a 1° longitude x 1°
latitude global grid with $N_z = 12$ vertical levels (every 1 km from 0 to 12 km, and 1 layer above 12 km).
Furthermore, the altitude of the 2 PVU potential vorticity level above or below the flight track is extracted from
the wind and temperature fields, in order to locate the CO observations above or below the dynamical tropopause
according to the approach of Thouret et al. (2006).
**3.2 Emission inventories from the ECCAD project**
The main goal of the Emissions of atmospheric Compounds & Compilation of Ancillary Data (ECCAD) project
(Granier et al., 2012) is to provide scientific and policy users with datasets of surface emissions of atmospheric





compounds and ancillary data, i.e. data required for estimating or quantifying surface emissions. All the emission
inventories and ancillary data provided by ECCAD are published in the scientific literature.
For the current study, we selected five CO emission inventories. Four of them are available at global scale
(MACCity and EDGAR v4.2 for anthropogenic; GFED 4 and GFAS v1.2 -GFAS v1.0 for 2002- for fires) from
the ECCAD database and cover most of the IAGOS CO database presented here (2002 - 2013). The global scale
inventories have a $0.1° \times 0.1°$ to $0.5° \times 0.5°$ horizontal resolution. They are provided with daily, monthly or
yearly time resolution. They are listed in Table 2 along with the references describing them. The four global
inventories are used to study the model's performance and sensitivity in Sect. 5.
To further test the sensitivity to the emission inventories, we also used one regional inventory, which is expected
to provide a better representation of emissions in its region of interest than generic global inventories. For
biomass burning, the International Consortium for Atmospheric Research on Transport and Transformation
(ICARTT) campaign's North American emissions inventory developed by Turquety et al. (2007) for the summer
of 2004 and provided at $1° \times 1°$ horizontal resolution was tested. It combines daily area burned data from forest
services with the satellite data used by global inventories, and uses a specific vegetation database, including
burning of peat lands which represent a significant contribution to the total emissions.
**3.3 Coupling transport output with CO emissions**
Calculating the recent contributions $C(i)$ (kg m$^{-3}$) of CO emissions for every one of the $i$ model's initialization
points along the flight tracks requires three kinds of data:
• the residence time $T_R$ (in seconds, gridded with $N_x = 360$ by $N_y = 180$ horizontal points, $N_z = 12$ vertical

levels, $N_t = 20$ days) from backward transport described in Sect. 3.1,

• CO surface emissions $E_{CO}$ ($N_x, N_y, N_t$) (in kg CO / m$^2$ / s)
• the injection profile $Inj(z)$ defining the fraction of pollutants diluted in the different vertical levels (with

$\Delta z$ being the thickness, in meters) just after emissions:


(Eq. 1) $$C(i) = \sum_{t=1}^{Nt} \sum_{y=1}^{Ny} \sum_{x=1}^{Nx} \sum_{z=1}^{Nz} Inj(z) \frac{T_R(x, y, z, t, i) E_{CO}(x, y, t)}{\Delta z(z)}$$

In the case of anthropogenic emissions, CO is simply emitted into the first vertical layer of the residence time
grid ($\Delta z = 1000$m).

For biomass burning emissions, in the tropics and mid latitudes regions, the lifting of biomass burning plumes is
usually due to small and large scale dynamical processes, such as turbulence in the boundary layer, deep
convection and frontal systems, which are usually represented by global meteorological models.  At higher
latitudes, however, boreal fires can also be associated with pyro-convection and quick injection above the
planetary boundary layer. Pyro-convection plume dynamics are often associated with small-scale processes that
are not represented in global meteorological data and emission inventories (Paugam et al 2016). In order to
characterize the effect of these processes, we implemented three methodologies to parameterize biomass
injection height:





- the first one (named DENTENER) depends only on the latitude and uses constant homogeneous injection profiles as defined by Dentener et al. (2006) ), i.e. 0-1 km for the tropics [30S-30N] (see green line in Fig 2), 0-2 km for the mid-latitudes [60S-30S, 30N-60N] (see blue line in Fig. 2) and 0-6 km for the boreal regions [90S-60S, 60N-90N ] (not shown in Fig. 2).

- the second named MIXED uses the same injection profiles as in DENTENER for the tropics and mid-latitudes, but for the boreal forest, injection profiles are deduced from a lookup table computed with the plume rise model PRMv2 presented in Paugam et al. (2015). Using PRMv2 runs for all fires from different years of the Northern-American MODIS archive, three daily Fire Radiative Power (FRP) classes (under 10 TJ/day, between 10 and 100 TJ/day, and over 100 TJ/day) were used to identify three distinct injection height profiles (see brown, red, and black lines in Fig. 2). Although PRMv2 reflects both effects of the fire intensity through the input of FRP and active fire size and effects of the local atmospheric profile, here for sake of simplicity only FRP is used to classify the injection profile. Furthermore, when applied to the IAGOS data set, the MIXED method uses equivalent daily FRP estimated from the emitted CO fluxes given by the emission inventories as described in Kaiser et al. (2012)

- the third method named hereafter APT uses homogeneous profile defined by the daily plume top altitude as estimated for each 0.1x0.1 pixel of the GFAS v1.2 inventory available for 2003 to 2013 (Rémy et al. 2016, and http://www.gmes-atmosphere.eu/oper_info/global_nrt_data_access/gfas_ftp/). As in the MIXED method, GFAS v1.2 is using the plume model PRMV2 from Paugam et al. (2015), but here the model is run globally for every assimilated GFAS-FRP pixel.

### 3.4 Automatic detection of CO anomalies

For individual measurement cases, plumes of pollution can most of the time be identified by the human eye using the observed CO mixing ratio time series or the CO vertical profiles. However, this is not feasible for a database of tens of thousands of observation flights. In order to create statistics of the model's performance, we need to systematically identify observed pollution plumes in the IAGOS database. The methodology to do this is based on what has been previously done for the detection of layers in the MOZAIC database (Newell et al., 1999; Thouret et al., 2000), along with more recent calculations of the CO background and CO percentiles define for different regions along the IAGOS data set (Gressent et al., 2014). An example demonstrating the procedure, which is described below, is shown in Fig. 3.

In a first step, the measurement time series along the flight track (number of measurements $n_{TOT}$) is separated into three parts:

1. Ascent and descent vertical profiles ($n_{VP}$) in the PBL (altitudes ranging from the ground to 2 km) and in the free troposphere (from 2 km to the top altitude of the vertical profiles),

2. measurements at cruising altitude in the upper troposphere ($n_{UT}$),

3. measurements in the lower stratosphere ($n_{LS}$)

such that $n_{TOT} = n_{VP} + n_{UT} + n_{LS}$



where $n_{VP}$, $n_{UT}$ and $n_{LS}$ are the number of measurements along tropospheric ascents and descents, and in the upper
troposphere and lower stratosphere, respectively. A range of altitudes from the surface to a top altitude identifies
vertical profiles. The top altitude is 75 hPa above the 2 pvu dynamical tropopause (Thouret at al., 2006) when
the aircraft reaches/leaves cruising altitude (during ascent/descent). The PV is taken from the ECMWF
operational analyses and evaluated at the aircraft position by FLEXPART. Observations made during the cruise
phase are flagged as upper tropospheric if the aircraft is below the 2 pvu dynamical tropopause. If not,
observations are considered as stratospheric and then are ignored in the rest of the paper. Although CO
contributions are calculated also in the stratosphere, the present study focuses on tropospheric pollution only.

In a second step, the CO background mixing ratio is determined for each tropospheric part ($C_{VP\_back}$ and $C_{UT\_back}$,
see Fig. 3 for illustration) for the tropospheric vertical profiles and for the upper troposphere respectively. For
tropospheric vertical profiles, the linear regression of CO mixing ratio versus altitude is calculated from 2 km to
the top of the vertical profiles, to account for the usual decrease of background CO with altitude. Data below
2 km are not used because high CO mixing ratios caused by fresh emissions are usually observed close to surface
over continents. The slope $a$ (in ppb m$^{-1}$) of the linear regression is used to determine the background so that
$C_{VP\_back} = aZ$. The background is removed from the $C_{VP}$ tropospheric vertical profiles mixing ratio to obtain a
residual CO mixing ratio $C^{R}_{VP}$ (Eq. 2).

(Eq. 2): $C^{R}_{VP} = C_{VP} - C_{VP\_back}$ ,


For the upper troposphere, the CO background mixing ratio ($C_{UT\_back}$) is determined using seasonal median
values (over the entire IAGOS database) for the different regions of Figure 4. Note that this approach was not
feasible for vertical profiles as for most of the visited airports there are not enough data to establish seasonal
vertical profiles. As for the profiles, background values are subtracted from the UT data to obtain residual $C^{R}_{UT}$
(Eq. 3):

(Eq. 3): $C^{R}_{UT} = C_{UT} - C_{UT\_back}$


In a third step, CO anomalies $C^{A}$ are determined for tropospheric vertical profiles ($C^{A}_{VP}$) and in the upper
troposphere ($C^{A}_{UT}$). Residual $C^{R}_{VP}$ and $C^{R}_{UT}$ values are flagged as CO anomalies when these values exceed the
third quartile (Q3) of the residual mixing ratio $C^{R}_{VP}(Q3)$ for vertical profiles, or the third quartile of the residual
seasonal values $C^{R}_{UT\_season}(Q3)$ in the different regions (Fig. 4) for the UT. Note that $C^{R}_{VP}(Q3)$ or $C^{R}_{UT\_season}(Q3)$
needs to be higher than 5 ppb (the accuracy of the CO instrument; Nédélec et al., 2015) in order to consider an
anomaly:

(Eq. 4): $C^{A}_{VP} = C^{R}_{VP}$   if   $C^{R}_{VP} > C^{R}_{VP}(Q3)$

(Eq. 5):   $C^{A}_{UT} = C^{R}_{UT}$ if $C^{R}_{UT} > C^{R}_{UT\_season}(Q3)$

In the examples shown in Fig. 3a and Fig. 3b, the red line represents CO anomalies.
With this algorithm CO plumes are automatically detected in the entire IAGOS database. For each identified
plume, minimum and maximum values of the date, latitude, longitude and altitude, as well as the CO mean and
maximum mixing ratio, are archived. These values are used for comparison with modeled CO values.



**4. Selected case studies to evaluate CO emission inventories and SOFT-IO's performance**

As described in Sect. 2, a number of case studies documented in the literature were selected from the IAGOS database in order to get a first impression of the model's performance. These case studies have been chosen to represent the different pollution situations that are often encountered in the troposphere in terms of emissions (anthropogenic or biomass burning) and transport (at regional or synoptic scale, pyro-convection, deep convection, frontal systems). Systematic evaluation of the model performance against emission inventories will be presented in Sect. 5.

**4.1 Anthropogenic emission inventories**

Among the case studies listed in Table 1, four were selected in order to illustrate the evaluation of the inventories used for anthropogenic emissions:

- Landing profiles over Hong Kong from 19th of July and 22nd of October 2005 were selected in order to investigate specifically Asian anthropogenic emissions.
- During the 10th of March 2002 Frankfurt–Denver and 27th of November 2002 Dallas–Frankfurt flights, IAGOS instruments observed enhanced CO plumes in the North Atlantic upper troposphere, also linked to anthropogenic emissions.

Figure 5a shows the observed (black line) and simulated (colored lines) CO mixing ratios above Hong Kong during 22nd of October 2005. Note that background is not simulated but estimated from the observations as described in Sect3.4 (blue line, $C_{VP\_back}$). The dashed blue line represents the residual CO mixing ratio $C^R_{VP}$. Observations show little variability in the free troposphere down to around 3 km. Strong pollution is observed below, with + 300 ppb enhancement over the background on average between 0 and 3 km. Note that we do not discuss CO enhancement above 3 km.

In agreement with $C^R_{VP}$, SOFT-IO simulates a strong CO enhancement in the lowest 3 km of the profile, caused by fresh emissions. However, the simulated enhancement is less strong than the observed one, a feature that is typical for this region, as we shall see later.

In addition to the CO mixing ratio, SOFT-IO calculates CO source contributions and geographical origins of the modeled CO, respectively displayed in Fig. 5b and Fig. 5c (using the methodology described in Sec. 3.4) and using here MACCity and GFAS v1.2 as example. For the geographical origin we use the same 14 regions as defined for the GFED emissions (http://www.globalfiredata.org/data.html). Note that only the average of the calculated CO is displayed for each anomaly (0-3km; 3.5-6km) in Fig. 5b and Fig. 5c.

Colored lines in Fig. 5a show the calculated CO using anthropogenic sources described by the two inventories selected in Sect. 3.2, MACCity (green line) and EDGARv4.2 (yellow line), along the flight track. In both cases, biomass burning emissions are described by GFASv1.2. Emissions from fires have negligible influence (less than 3%) on this pollution event as depicted in Fig. 5b.

In the two simulations, the calculated CO mixing ratio is below 50 ppb in the free troposphere, as we do not simulate background concentrations with SOFT-IO. CO enhancement around 4 to 6 km is overestimated by SOFT-IO. CO above 6 km is not considered as an anomaly, as $C^R_{UT} < C^R_{UT\_season}(Q3)$. Simulated mixing ratios in the 0-2 km polluted layer are almost homogeneous, with values around 280 ppb using MACCity and around 160 ppb using EDGARv4.2. They are attributed to anthropogenic emissions (more than 97% of the simulated CO)





originating mostly from Central Asia with around 95% influence. In this regard, the CO simulated using
MACCity is in better agreement with the observed CO than the one obtained using EDGARv4.2. Indeed, using
MACCity, simulated CO reaches 90% of the observed enhancement (+ 300 ppb on average) over the background
(around 100 ppb), while for EDGARv4.2 the corresponding value is only 53%, indicating strong underestimation
of this event. The difference in the calculated CO using these two inventories is also consistent with the results
of Granier et al. (2011) who showed strong discrepancies in the Asian anthropogenic emissions in different
inventories.

Figure 6a shows the CO measurements at cruising altitude during a transatlantic flight between Frankfurt and
Denver on 10th of March 2002. The dashed blue line represents the residual CO $C^R_{UT}$. Observations indicate that
the aircraft encountered several polluted air masses with CO mixing ratios above 110 to 120 ppb, which are the
seasonal median CO values in the two regions visited by the aircraft, obtained from the IAGOS database (see
Gressent et al., 2014). Three pollution plumes are measured:
• around 100°W (around +10 ppb of CO enhancement on average): plume 1
• between 80°W and 50°W (+30 ppb of CO enhancement on average): plume 2
• between 0° and 10°E (+40 ppb of CO enhancement on average): plume 3.
These polluted air masses are surrounded by stratospheric air masses with CO values lower than 80-90 ppb. As
polluted air masses were sampled at an altitude of around 10 km, they are expected to be due to long-range
transport of pollutants.
The calculated CO is shown in Fig. 6a using MACCity (green line), EDGARv4.2 (yellow line) for anthropogenic
emissions and GFASv1.0 for biomass burning emissions. SOFT-IO estimates that these plumes are mostly
anthropogenic (representing 77% to 93% of the total simulated CO, Fig. 6b). Pollution mostly originates from
Central and South-East Asia, with strong contribution from North America (Fig. 6c) for plume 3.
SOFT-IO correctly locates the three observed polluted air masses with the two anthropogenic inventories. CO is
also correctly calculated using MACCity, with almost the same mixing ratios on average as the observed
enhancements in the three plumes. Only 2/3 of the observed enhancements are simulated using EDGARv4.2,
except for plume 1 with better results. We have already seen in the previous case study that emissions in Asia
may be underestimated, especially in the EDGARv4.2 inventory.
Similar comparisons were performed in the four case studies selected to estimate and validate the anthropogenic
emission inventories coupled with the FLEXPART model. Results are summarized in Table 3. For three of the
cases, SOFT-IO simulations showed a better agreement with observations when using MACCity than when
using EDGARv4.2. In the fourth case both inventories performed equally well. One reason for the better
performance of MACCity is the fact that it provides monthly information (Table 2).

**4.2 Biomass burning emission inventories**
In order to evaluate and choose biomass burning emission inventories, we have selected eleven case studies with
fire-induced plumes (Table 1). Seven of them focused on North-American biomass burning plumes observed in
the free troposphere above Europe (flights on 30th of June, 22nd and 23rd of July 2004) and in the upper
troposphere/lower stratosphere above the North Atlantic (29th of June 2004) (e.g. Elguindi et al., 2010; Cammas



et al., 2009). Two are related to the fires over Western Europe during the 2003 heat wave (Tressol et al. 2008).
The two last ones, on the 30th and 31st of July 2008, focused on biomass burning plumes observed in the ITCZ
region above Africa as described in a previous study (Sauvage et al., 2007a).
The three datasets selected to represent biomass burning emissions are based on different approaches: GFAS
v1.2 (Kaiser et al., 2012) and GFED 4 emissions (Giglio et al., 2013) are calculated daily. GFAS v1.2 presents
higher spatial resolution. The ICARTT campaign inventory (Turquety et al., 2007) was specifically designed for
North-American fires during the summer of 2004 with additional input from local forest services.
Figure 7a illustrates the calculated CO contributions for the different fire emission inventories for one of the case
studies, on 22nd of July 2004 above Paris. The observations (black line) show high levels of CO in an air mass in
the free troposphere between 3 and 6 km, with mixing ratios 140 ppb above the background (blue line) deduced
from measurements. This pollution was attributed to long-range transport of biomass burning emission in North
America by Elguindi et al. (2010). Outside of the plume, the CO concentration decreases with altitude, from
around 150 ppb near the ground, to 100 ppb background in the upper free troposphere. This last value
corresponds to the median CO seasonal value deduced from the IAGOS database (Gressent at al., 2014). CO is
not considered as an anomaly near the ground as $C^R_{UT} < C^R_{UT\_season}$(Q3).
SOFT-IO simulations were performed for this case using MACCity to represent anthropogenic emissions, and
GFAS v1.2 (green line), GFED 4 (yellow line), or the ICARTT campaign inventory (red line). Fire vertical
injection is realized using the MIXED approach for the three biomass burning inventories, in order to only
evaluate the impact of choosing different emission inventories. In the three simulations, contributions show two
peaks, one near the ground that is half due to local anthropogenic emissions and half due to contributions from
North American biomass burning and thus not considered in this discussion.
The second more intense peak, simulated in the free troposphere where the enhanced CO air masses were
sampled, is mostly caused by biomass burning emissions (87% of the total calculated CO, Fig. 7b), originating
from North-America (99% of the total enhanced CO). When calculated using the ICARTT campaign inventory,
the simulated CO enhancement reaches over 150 ppb, which is 10 ppb higher than the observed mixing ratio
above the background (+140 ppb), but only for the upper part of the plume.
When using global inventories, the simulated contribution peak reaches 70 ppb using GFASv1.2 and 100 ppb
using GFED4, which appears to underestimate the measured enhancement (+140 ppb) by up to 50% to 70%
respectively. This comparison demonstrates the large uncertainty in simulated CO caused by the emission
inventories, both in the case of biomass burning or anthropogenic emissions. For that reason we aim to provide
simulations with different global and regional inventories in for the IAGOS data set.
As the ICARTT campaign inventory was created using local observations in addition to satellite products, the
large difference in the simulated CO compared to the other inventories may in part be due to different
quantification of the total area burned (for GFED, GFAS using the FRP as constraint). Turquety et al. (2007)
also discussed the importance of peat land burning during that summer. They estimated that they contributed
more than a third of total CO emissions (11 Tg of the 30 Tg emitted during summer 2004).

Figure 8a shows CO mixing ratios as a function of latitude for a flight from Windhoek (Namibia) to Frankfurt
(Germany) in July 2008. Observations indicate that the aircraft flew through polluted air masses around the
equator (10°S to 10°N), with +100 (+125) ppb of CO on average (at the most) above the 90 ppb background



deduced from seasonal IAGOS mixing ratios over this region. Such CO enhancements have been attributed to
regional fires injected through ITCZ convection (Sauvage et al., 2007b).
The SOFT-IO simulations (colored lines in Fig. 8a) link these air masses mostly to recent biomass burning
(responsible for 68% of the total simulated CO, Fig. 8b) in South Africa (Fig. 8c). The calculated CO shows
similar features both with GFED4 (yellow line) and GFASv1.2 (green line). The simulation also captures well
the intensity variations of the different peaks: maximum values around the equator, lower ones south and north
of the equator. The most intense simulated CO enhancement around the equator fits the observed CO
enhancement of +125 ppb better when using GFED4 (90 ppb) than when using GFASv1.2 (75 ppb). However
the comparison also reveals an underestimation of the CO anomaly's amplitude by around 10 ppb to 25 ppb on
average by SOFT-IO. The model is thus only able to reproduce 75% to 90% of the peak concentrations on
average. Stroppiana et al. (2010) indeed showed that there are strong uncertainties in the fire emission
inventories over Africa (164 to 367 Tg CO per year).

## 440 5 Statistical evaluation of the modeled CO enhancements in pollution plumes

In this section, we present a statistical validation of the SOFT-IO calculations based on the entire IAGOS CO
data base (2003-2013). The ability of SOFT-IO in simulating CO anomalies is evaluated compared to in situ
measurements in terms of:
• spatial and temporal frequency of the plumes
• mixing ratio enhancements in the plumes
To achieve this, SOFT-IO performances are investigated over different periods of IAGOS measurements
depending on the emission inventory used. Three of the four global inventories selected previously (MACCity,
GFAS v1.2, GFED4) are available between 2003 and 2013. EDGAR v4.2 ends in 2008.  In the following
sections (Sect.5.1 and 5.2), we discuss in detail the results obtained with MACCity and GFAS v1.2 between
2003 and 2013. Other emission inventory combinations are discussed in Sect. 5.3 when investigating SOFT-IO
sensitivity to input parameters.

### 452 5.1 Detection frequency of the observed plumes with SOFT-IO

The ability of SOFT-IO to reproduce CO enhancements was investigated using CO plumes obtained applying the
methodology described in Sect. 3.4 on all flights of the IAGOS database between 2003 and 2013. The frequency
of simulated plumes that coincide with the observed $C^A$ anomalies is then calculated. Simulated plumes are
considered when matching in time and space the observed plumes, while modeled CO is on average higher than
5 ppb within the plume. Note that at this stage, we do not consider the intensity of the plumes.
The resulting detection rates are presented in Fig. 9 for eight of the eleven regions shown in Fig. 4. Statistics are
presented separately for three altitude levels (Lower Troposphere 0-2 km, Middle Troposphere 2-8 km and
Upper Troposphere > 8 km). Figure 9 shows that SOFT-IO performance in detecting plumes is very good and
not strongly altitude or region-dependent. In the three layers (LT, MT and UT), detection rates are higher than
95% and even close to 100% in the LT where CO anomalies are often related to short-range transport. Detection
frequency slightly decreases in the MT and the UT where CO modeling accuracy suffers from larger errors in
vertical and horizontal transport. On the contrary CO anomalies in the LT are most of the times related to short-
range transport of local pollution, which are well represented in SOFT-IO. For four regions we found less good





results: South America MT and UT, Africa MT and North Asia UT but with still high detection frequency (82%
to 85%). Note that only relatively few plumes (313 to 3761) were sampled by the IAGOS aircraft fleet in these
regions.

**5.2 Intensity of the simulated plumes**
The second objective of SOFT-IO is to accurately simulate the intensity of the observed CO anomalies. Fig. 10a
displays the bias between the means of the observed and modeled plumes for the regions sampled by IAGOS and
in the three vertical layers (LT, MT and UT). As explained above this bias is calculated for the 2003-2013 period
and using both anthropogenic emission from MACCity and biomass burning emissions from GFAS v1.2 and the
plume detection methodology described in Sect. 3.4.
The most documented regions presenting CO polluted plumes (Europe, North America, Africa, North Atlantic
UT, Central Asia MT and UT, South America, South Asia UT) present low biases (lower than ± 5 ppb, and up to
± 10 ppb for Central Asia MT, South America UT), which demonstrate a high skill of SOFT-IO.
Over several other regions with less frequent IAGOS flights, however, biases are higher, around ±10-15 ppb for
Africa UT and South Asia MT; around ± 25-50 ppb for Central Asia LT, South Asia LT and North Asia UT.
Except for the last region, the highest biases are found in the Asian lower troposphere, suggesting
misrepresentation of local emissions. Indeed there is a rapid increase of emissions in this large area (Tanimoto et
al., 2009) associated with high discrepancies between different emission inventories (Wang et al., 2013; Stein et
al., 2014) and underestimated emissions (Zhang et a., 2015).
It is important to note that the biases remain of the same order (±10-15 ppb) when comparing the first (Q1),
second (Q2) and third (Q3) quartiles of the CO anomalies observed and modeled within most of the regions (Fig.
10b). This confirms the good capacity of the SOFT-IO software in reproducing the CO mixing ratios anomaly in
most of the observed pollution plumes.
Differences become much larger when considering outlier values of CO anomalies (lower and upper whiskers, ±
2.7σ or 99.3%, Fig. 10b), which means for exceptional events of very low and very high CO enhancements
(accounting for 1.4% of the CO plumes), with biases from ± 10 ppb to ± 50 ppb for most of the regions. Higher
discrepancies are found in the lower and the upper troposphere and can reach ±50 to ±200 ppb in two specific
regions (North Asia UT and South Asia LT) for these extreme CO anomalies. Note that North Asia UT and
South Asia LT present respectively extreme pollution events related to pyro-convection (Nédélec et al., 2005) for
the first region, and to strong anthropogenic surface emissions (Zhang et al., 2012) for the second one. It may
suggest that the model fails to correctly reproduce the transport for some specific but rare events of pyro-
convection.
When looking at the origin of the different CO anomalies (Fig. 10c), most of them are dominated by
anthropogenic emissions, which account for more than 70% of the contributions on average, except for South
America and Africa, which are strongly influenced by biomass burning (Sauvage et al. 2005, 2007c; Yamasoe et
al., 2014). Discussing origins of the CO anomalies in detail is out of the scope of this study, but gives here some
information on the model performance. It is interesting to note that two of the three regions most influenced by
anthropogenic emissions, South Asia LT and Central Asia LT, with more than 90% of the enhanced CO coming
from anthropogenic emissions, are the highest biased regions compared to observations. This is not the case for




Europe LT for example, which also has a high anthropogenic influence. As stated before, anthropogenic
emissions in Asia are more uncertain than elsewhere (Stein et al., 2014).

In order to go a step further in the evaluation of SOFT-IO in reproducing CO anomalies mixing ratios, Fig. 11
displays the monthly mean time series of the observed (black line) and calculated (blue line) CO anomalies in
three vertical layers (LT, MT and UT). This graph provides higher temporal resolution of the anomalies. CO
polluted plumes are displayed here using MACCity and GFAS v1.2 over the 2003-2013 periods and for the two
regions with the largest number of observed CO anomalies, Europe and North America.
It is worth noting the good ability of SOFT-IO in quantitatively reproducing the CO enhancements observed by
IAGOS. This is especially noticeable in the LT and UT, with similar CO mixing ratios observed and modeled
during the entire period and within the standard deviation. However, the amplitude of the seasonal cycle of CO
maxima is highly underestimated (-100%) after January 2009 in the European LT, where anthropogenic sources
are predominant with more than 90% influence (Fig. 10c). This suggests misrepresentation of anthropogenic
emissions in Europe after the year 2009 (Stein et al., 2014).
In the middle troposphere (2-8 km), the CO plumes are systematically overestimated by SOFT-IO by 50% to
100% compared to the observations. This might be related to different reasons:
• the chosen methodology of the CO plume enhancements detection for those altitudes (described in Sect.
3.4), which may lead to a large number of plumes with small CO enhancements, which are difficult to
simulate. This could be due to the difficulty in defining a realistic CO background in the middle
troposphere.
• the source-receptor transport which may be more difficult to simulate between 2-8 km than in the LT
where receptors are close to sources; or than in the UT where most of the plumes are related to
convection detrainment better represented in the models than MT detrainment which might be less
intense.
• The frequency of the IAGOS observations which is lower in the MT than in the UT.
Correlation coefficients between simulated and observed plumes are highest in the LT (0.56 to 0.79) and lower
(0.30 to 0.46) in the MT and in the UT, suggesting some difficulties for the model in lifting up pollution from the
surface to the UT.

**5.3 Sensitivity of SOFT-IO to input parameters**

Different factors influence the ability of SOFT-IO to correctly reproduce CO pollution plumes. Among them,
transport parameterizations (related to convection, turbulence, etc) are not evaluated in this study as they are
inherent of the FLEXPART model. In this section, the model sensitivity to the chosen emission inventory is
evaluated. For this, a set of sensitivity studies is performed to investigate different configurations of the emission
inventories :
• type of inventory: MACCity, EDGAR for anthropogenic, GFED4, GFAS v1.2 or ICARTT for biomass
burning
• biomass burning injection heights: DENTENER, MIXED or APT approach (detailed in Sect. 3.3).





SOFT-IO performances are then investigated using Taylor diagrams (Taylor et al. 2001). The methodology
(choice of regions, vertical layers, sampling periods) is similar to the one used to analyze the ability of the model
to correctly reproduce the frequency and the intensity of the CO plumes with MACCity and GFAS (Sect.5.1 and
Sec5.2).
**5.3.1 Anthropogenic emission inventories**
Sensitivity of SOFT-IO to anthropogenic emissions is investigated between 2002 and 2008, using GFAS with
MACCity or EDGARv4.2. Fig. 12a presents a Taylor diagram for the two configurations (dots for MACCity,
crosses for EDGAR) for the regions and for the vertical layers described previously (Sect. 5.1 and Sect. 5.2),
while Fig. 12b represents the mean bias between each model configuration and the IAGOS observations.
As already seen in Sect. 4.1 for the case studies chosen to investigate anthropogenic emissions, slightly better
results seem to be obtained with MACCity. The Taylor diagram shows for most of the regions higher
correlations and lower biases in this case. These results are not surprising, as MACCity (Granier et al., 2011) is a
more recent inventory compared to EDGARv4.2 (Janssens-Maenhout et al., 2010), and expected to better
represent anthropogenic emissions. However the differences between the two inventories are most of the time
very low, as global emission inventories tend to be quite similar.
Regionally, however, results with EDGARv4.2 can be better, such as over South Asia LT and MT, Central Asia
MT and North Asia UT. This supports our choice of maintaining several different inventories in SOFT-IO.
**5.3.2 Biomass burning emissions**
We first investigate the sensitivity of SOFT-IO to the type of biomass burning inventory, using MACCity with
GFAS v1.2 or GFED 4 (2003-2013), using the same MIXED methodology for vertical injection of emissions
(Fig. 2). As for anthropogenic emissions, Fig. 13 represents the Taylor diagram and averaged biases for the
different configurations.
Performances (correlations, standard deviations and biases) are very similar for both biomass burning
inventories, with smaller differences compared to anthropogenic inventories. Even for regions dominated by
biomass burning such as Africa or South America as depicted previously (Fig. 11c), the sensitivity of the SOFT-
IO performance to the type of global fire inventory is below 5 ppb.

Based on case studies, we discussed in Sect. 4.2 the comparison of CO contributions modeled using regional fire
emission inventories. It resulted in a better representation of biomass burning plumes using the specifically
designed campaign inventory than using the global inventories (Table 4). However, there is no clear evidence of
this result when investigating the model performances during the whole summer 2008. On contrary to Sect. 4.2,
it is hard to conclude of systematic better results using the ICARTT inventory.  While simulations (not shown)
give better results for a few specific events of very high CO using ICARTT, similarly good results are obtained
when using GFASv1.2 or GFED4 for most other cases. It is worth noting that IAGOS samples biomass burning
plumes far from ICARTT sources, after dispersion and diffusion during transport in the atmosphere. Besides,
few boreal fire plumes (that would be better represented using ICARTT), are sampled by the IAGOS program.




Secondly, we investigate the influence of the vertical injection scheme for the biomass burning emissions, using
the three methodologies for determining injection heights described in Sect. 3.3. Sensitivity tests (Fig. 13c and
Fig 13d) demonstrate a small influence of the injection scheme on the simulated plumes. The largest influence is
found over North Asia UT, where pyro-convection has been highlighted in the IAGOS observations (Nédélec et
al., 2005), with however less than 5 ppb difference between the different schemes. More generally, small vertical
injection influence is probably due to too few cases where boreal fire emissions are injected outside the PBL by
pyro-convection, as shown in the Paugam et al. (2016) study, combined with a too low sampling frequency of
boreal fire plumes by IAGOS.

**6 Conclusions**

Analyzing long term in situ observations of trace gases can be difficult without a priori knowledge of the
processes driving their distribution and seasonal/regional variability, like transport and photochemistry. This is
particularly the case for the extensive IAGOS database, which provides a large number of aircraft-based in-situ
observations (more than 51000 flights so far) distributed on a global scale, and with no a priori sampling
strategy, unlike dedicated field campaigns.

In order to help studying and analyzing such a large data set of in situ observations, we developed a system that
allows quantifying the origin of trace gases both in terms of geographical location as well as source type. The
SOFT-IO module (https://doi.org/10.25326/2) is based on the FLEXPART particle dispersion model that is run
backward from each trace gas observation, and on different emission inventories (EDGAR v4.2, MACCity,
GFED 4, GFAS v1.2) than can be easily changed.

The main advantages of the SOFT-IO module are:
- Its flexibility. Source-receptor relationships pre-calculated with the FLEXPART particle dispersion
model can be coupled easily with different emission inventories, allowing each user to select model
results based on a range of different available emission inventories.
- CO calculation, which is computationally very efficient, can be repeated easily whenever updated
emission information becomes available without running again the FLEXPART model. It can also be
extended to a larger number of emission datasets, particularly when new inventories become available,
or for emission inventories inter-comparisons. It can also be extended to other species with similar or
longer lifetime as CO to study other type of pollution sources.
- High sensitivity of the SOFT-IO CO mixing ratios to source choice for very specific regions and case
studies, especially in the LT most of the time driven by local or regional emissions, may also help
improving emission inventories estimates through evaluation with a large database such as IAGOS one.
Indeed as it is based on a Lagrangian dispersion model, the tool presented here is able to reproduce
small-scale variations, which facilitates comparison to in situ observations. It can then be used to
validate emission inventories by confronting them to downwind observations of the atmospheric
composition, using large database of in situ observations of recent pollution.


- More generally SOFT-IO can be used in the future for any kind of atmospheric observations (e.g. ground based measurements, satellite instruments, aircraft campaigns) of passive tracers.

In this study SOFT-IO is applied to all IAGOS CO observations, using ECMWF operational meteorological analysis and 3-hour forecast fields and inventories of anthropogenic and biomass burning emissions available on the ECCAD portal. SOFT-IO outputs are evaluated first at the examples of case studies of anthropogenic and biomass burning pollution events. The evaluation is then extended statistically, for the entire 2003-2013 period, over 14 regions and 3 vertical layers of the troposphere.

The main results are the following:

- By calculating the contributions of recent emissions to the CO mixing ratio along the flight tracks, SOFT-IO identifies the source regions responsible for the observed pollution events, and is able to attribute such plumes to anthropogenic and/or biomass burning emissions.

- On average, SOFT-IO detects 95% of all observed CO plumes. In certain regions, detection frequency reaches almost 100%.

- SOFT-IO gives a good estimation of the CO mixing ratio enhancements for the majority of the regions and the vertical layers. In majority, the CO contribution is reproduced with a mean bias lower than 10-15 ppb, except for the measurements in the LT of Central and South Asia and in the UT of North Asia where emission inventories seems to be less accurate.

- CO anomalies calculated by SOFT-IO are very close to observations in the LT and UT where most of the IAGOS data are recorded. Agreement is lower in the MT, possibly because of numerous thinner plumes of lower intensity (maybe linked to the methodology of the plume selection).

- SOFT-IO has less skill in modeling CO in extreme plume enhancements with biases higher than 50 ppb.

In its current version, SOFT-IO is limited by different parameters, such as inherent parameterization of the Lagrangian model, but also by input of external parameters such as meteorological field analysis and emission inventories. Sensitivity analyses were then performed using different meteorological analysis and emissions inventories, and are summarized as follow:

- Model results were not very sensitive to the resolution of the meteorological input data. Increasing the resolution from 1 deg to 0.5 deg resulted only in minor improvements. On the other hand, using operational meteorological analysis allowed more accurate simulations than using ERA-Interim reanalysis data, perhaps related to the better vertical resolution of the former.

- Concerning anthropogenic emissions sensitivity tests, results display regional differences depending on the emission inventory choice. Slightly better results are obtained using MACCity.

- Model results were not sensitive to biomass burning global inventories, with good results using either GFED 4 or GFAS v1.2. However, a regional emission inventory shows better results for few individual cases with high CO enhancements. There is a low sensitivity to parameterizing the altitude of fire emission injection, probably because events of fires injected outside of the PBL are rare or because IAGOS does not frequently sample of such events



Using such CO calculations and partitioning makes it possible to link the trends in the atmospheric composition
with changes in the transport pathways and/or changes of the emissions.
SOFT-IO products will be made available through the IAGOS central database (http://iagos.sedoo.fr/#L4Place)
and are part of the ancillary products (https://doi.org/10.25326/3)




**Acknowledgements**

The authors would like to thanks ECCAD project for providing emission inventories. The authors acknowledge
the strong support of the European Commission, Airbus, and the Airlines (Lufthansa, Air-France, Austrian, Air
Namibia, Cathay Pacific, Iberia and China Airlines so far) who carry the MOZAIC or IAGOS equipment and
perform the maintenance since 1994. In its last 10 years of operation, MOZAIC has been funded by INSU-
CNRS (France), Météo-France, Université Paul Sabatier (Toulouse, France) and Research Center Jülich (FZJ,
Jülich, Germany). IAGOS has been additionally funded by the EU projects IAGOS-DS and IAGOS-ERI. The
MOZAIC-IAGOS database is supported by AERIS (CNES and INSU-CNRS). The former CNES-ETHER
program has funded this project.






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








| Date | Take-off | Landing | Used for choosing |
|---|---|---|---|
| **10 March 2002** | **Frankfurt** | **Denver** | **Anthropogenic emission inventories** |
| 27 November 2002 | Dallas | Frankfurt | Anthropogenic emission inventories |
| 4 June 2003 | Tokyo | Vienna | Fire injection heights (pyro-convection) |
| 6 August 2003 | Boston | Frankfurt | Fire injection heights |
| 9 August 2003 | Dubai | Frankfurt | Fire injection heights |
| 10 August 2003 | Frankfurt | Dallas | Fire injection heights |
| 29 June 2004 | Caracas | Frankfurt | Fire injection heights (pyro-convection) |
| 30 June 2004 | Frankfurt | Washington | Fire injection heights (pyro-convection) Fire inventories |
| 22 July 2004 | Frankfurt | Atlanta | Fire injection heights (pyro-convection) Fire inventories |
| **22 July 2004** | **Douala** | **Paris** | **Fire injection heights (pyro-convection) Fire inventories** |
| 23 July 2004 | Frankfurt | Atlanta | Fire injection heights (pyro-convection) Fire inventories |
| 19 July 2005 | München | Hong Kong | Anthropogenic emission inventories |
| **22 October 2005** | **München** | **Hong Kong** | **Anthropogenic emission inventories** |
| **30 July 2008** | **Windhoek** | **Frankfurt** | **Fire injection heights Fire emission inventories** |
| 31 July 2008 | Frankfurt | Windhoek | Fire injection heights Fire emission inventories |

**Table 1: Case studies used to define model settings. Cases studies discussed in the manuscript are in bold**











| Inventory | Temporal coverage | Horizontal resolution | Temporal resolution | Reference |
|---|---|---|---|---|
| | *Anthropogenic emissions* | | | |
| MACCity | 1960 – 2014 + | 0.5° x 0.5° | Monthly | *Granier et al. (2011)* |





| EDGAR v4.2 | 1970 - 2008 | 0.5° x 0.5° | Yearly | *Janssens-Maenhout et al. (2010)* |
|---|---|---|---|---|
| | *Biomass Burning emissions* | | | |
| GFED 4 | 1997 – 2017+ | 0.5° x 0.5° | Daily | *Giglio et al. (2013)* |
| GFAS v1.0 | 2002 | 0.5° x 0.5° | Daily | |
| GFAS v1.2 | 2003 – 2017 + | 0.1° x 0.1° | Daily | *Kaiser et al. (2012)* |
| ICARTT | 2004 | 1° x 1° | Daily | *Turquety et al. (2007)* |

**Table 2: List of emission inventories used in this study.**

| Flight | MACCity | EDGAR v4.2 |
|---|---|---|
| 10 March 2002 Frankfurt – Denver | + | |
| 27 November 2002 Dallas – Frankfurt | = | = |
| 19 July 2005 München - Hong Kong | + | |
| 22 October 2005 München - Hong Kong | + | |

**Table 3. Summary of optimal inventory (indicated by a plus sign) determined for representing anthropogenic**
**emissions for different case studies. Equal sign indicate that the case is non-conclusive.**

| Flight | GFED 4 | GFAS v1.2 | ICARTT |
|---|---|---|---|
| 29 June 2004 Caracas - Frankfurt | | | + |
| 30 June 2004 Frankfurt - Washington | | | + |
| 22 July 2004 Frankfurt - Atlanta | | | + |
| 22 July 2004 Douala - Paris | | | + |
| 23 July 2004 Frankfurt - Atlanta | | | + |
| 30 July 2008 Windhoek - Frankfurt | + | | N/A |
| 31 July 2008 Frankfurt - Windhoek | = | = | N/A |

**Table 4. Summary of optimal inventory (indicated by a plus sign) determined for representing fire emissions for**
**different case studies. Equal signs indicate that the case is non-conclusive. Note that the ICARTT inventory is only**
**available for summer 2004.**





















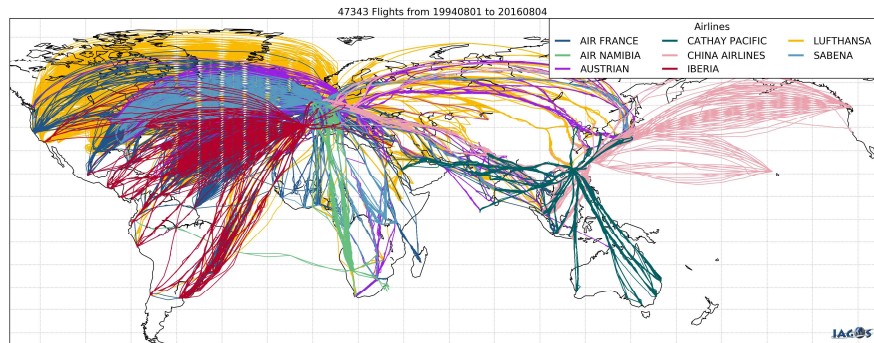

**Figure 1 : Map showing all flights performed by the IAGOS program**

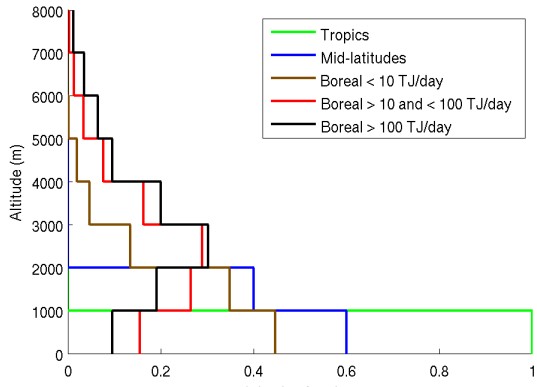

**Figure 2: Injection profiles used for biomass burning emissions for different regions (Tropics, Mid-latitudes, Boreal)**
**in the MIXED methodology.**





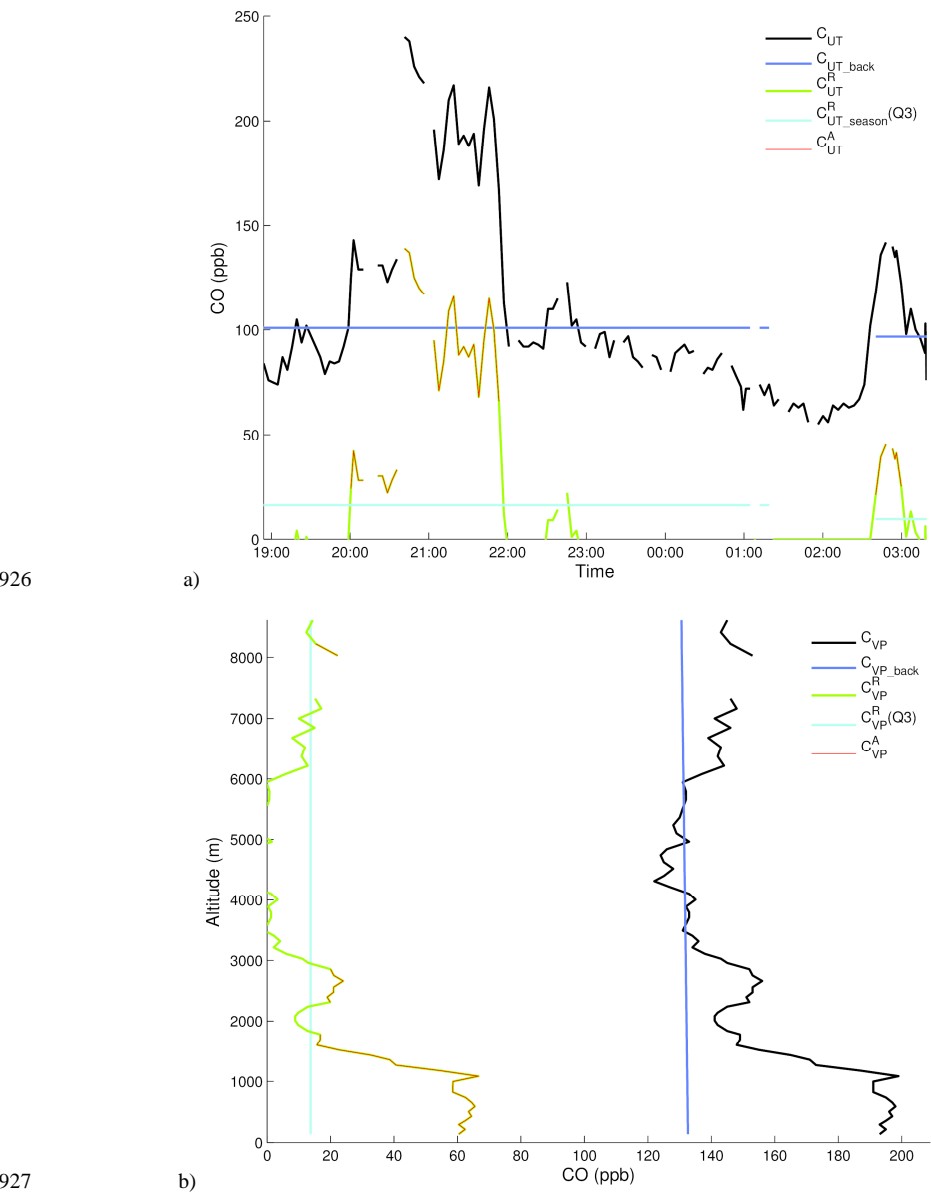

a)

b)

**Figure 3: Methodology used to extract CO anomalies along the flight track for (a) the cruise part of the flight and (b) during take off and landing. Further details are given in section 3.4.**



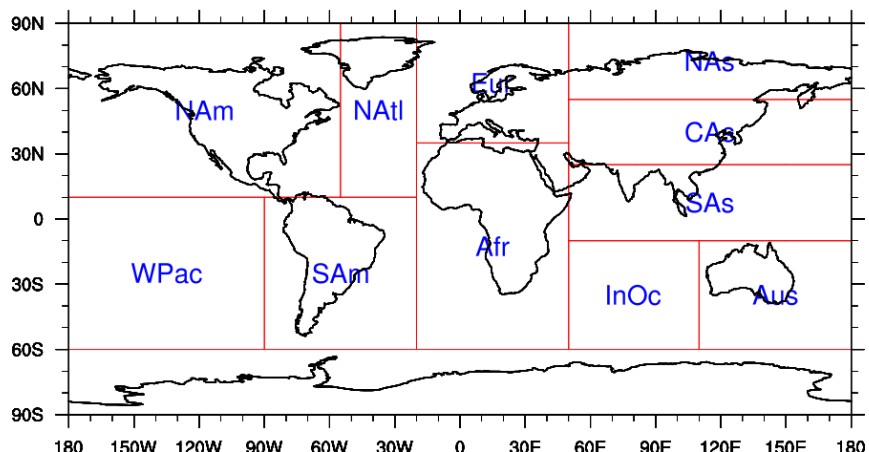

**Figure 4: Map of the defined regions used to sort IAGOS CO anomalies**



a)

b)





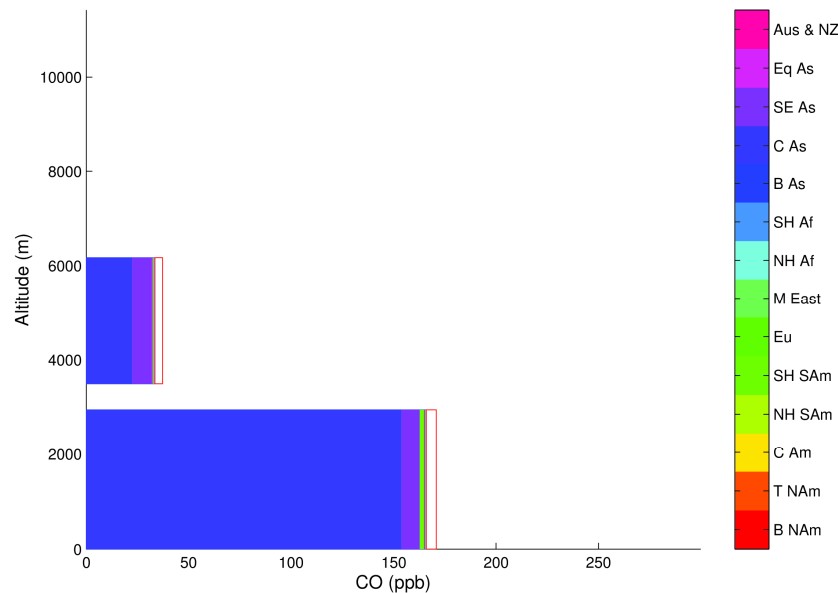

c)

**Figure 5: (a) Carbon monoxide profiles over Hong Kong during a MOZAIC-IAGOS flight landing on 22 October 2005. The black line indicates the observed CO profile while the blue line indicates the CO background deduced from the observations. Green and yellow lines indicate the simulated CO contributions using respectively MACCity and EDGARv4.2 for anthropogenic emissions, and using GFAS v1.2 for biomass burning emissions. Simulated CO is separated in (b) sources contribution (anthropogenic in blue, fires in red) and in (c) regional anthropogenic origins (14 regions defined for global emission inventory, http://www.globalfiredata.org/data.html; unshaded red square is for fire contribution), using MACCity and GFASv1.2.**

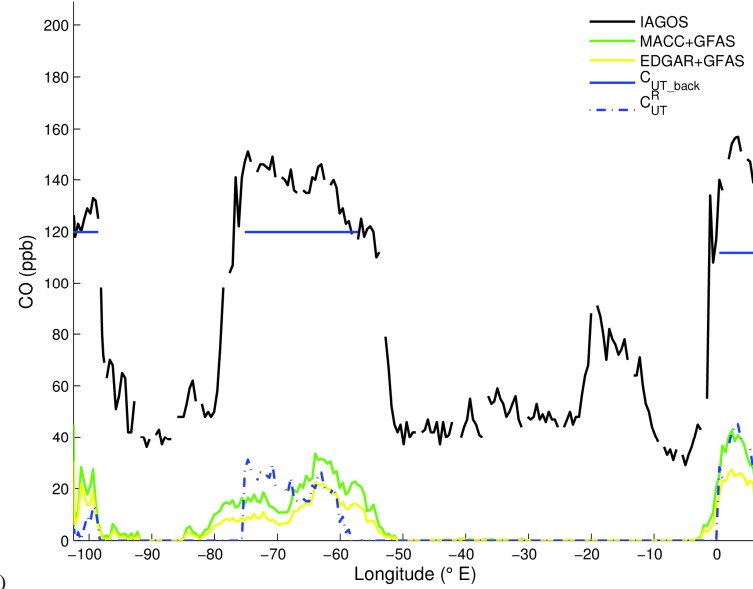

a)



b)

c)

**Figure 6: (a)** Carbon monoxide zonal profile during the 10 March 2002 MOZAIC-IAGOS flight from Frankfurt to Denver. The black line indicates the observed CO while the blue line indicates CO seasonal background in the UT deduced from the IAGOS data set. Green and yellow lines indicate the simulated contributions using respectively MACCity and EDGARv4.2 for anthropogenic emissions, and GFAS v1.0 for biomass burning emissions. Simulated CO is separated in **(b)** sources contribution (anthropogenic in blue, fires in red) and in **(c)** regional anthropogenic origins (14 regions defined for global emission inventory, http://www.globalfiredata.org/data.html; unshaded red square is for fire contribution), using MACCity and GFASv1.0.




a)

b)




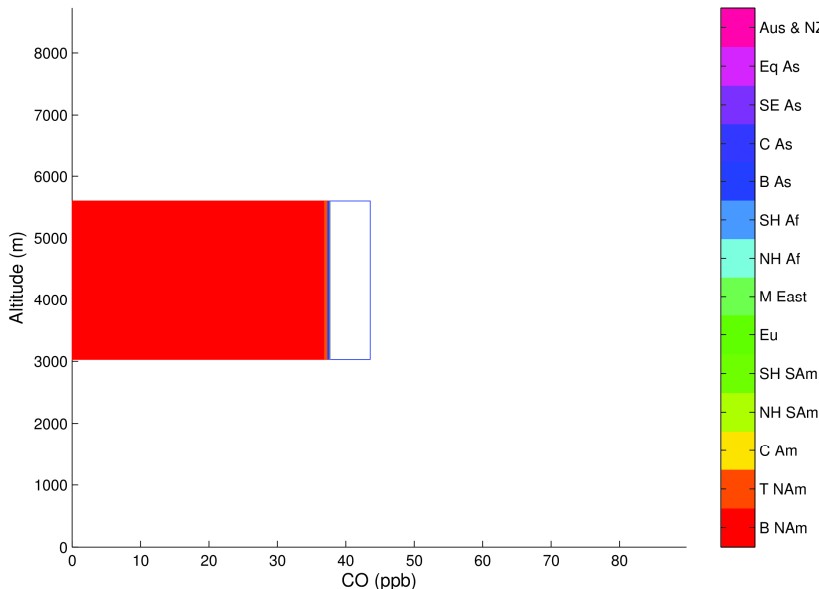

c)
**Figure 7 : (a) Carbon monoxide profiles over Paris during a MOZAIC-IAGOS flight landing on 22 July 2004. The**
**black line indicates the observed CO profile and the blue line indicates CO background deduced from the**
**observations. Green, yellow and red lines indicate the simulated contributions using respectively GFASv1.2, GFED4**
**and ICARTT for biomass burning emissions, with MACCity for anthropogenic emissions. Simulated CO is separated**
**in (b) sources contribution (anthropogenic in blue, fires in red) and in (c) regional biomass burning origins (14 regions**
**defined for global emission inventory, http://www.globalfiredata.org/data.html; unshaded blue square is for**
**anthropogenic contribution), using MACCity and GFASv1.2.**

a)





b)

c)

**Figure 8: (a) Carbon monoxide as a function of latitude during the 30 July 2008 MOZAIC-IAGOS flight from**
**Windhoek to Frankfurt. The black line indicates the observed CO, the blue line indicates the CO seasonal**
**background deduced from the IAGOS data set. Green and yellow lines indicate the simulated contributions using**
**MACCity for anthropogenic emissions, and respectively GFAS v1.2 and GFED4 for biomass burning emissions.**
**Simulated CO is separated in (b) sources contribution (anthropogenic in blue, fires in red) and in (c) regional biomass**
**burning origins (14 regions defined for global emission inventory, http://www.globalfiredata.org/data.html; unshaded**
**red square is for anthropogenic contribution), using MACCity and GFASv1.2.**




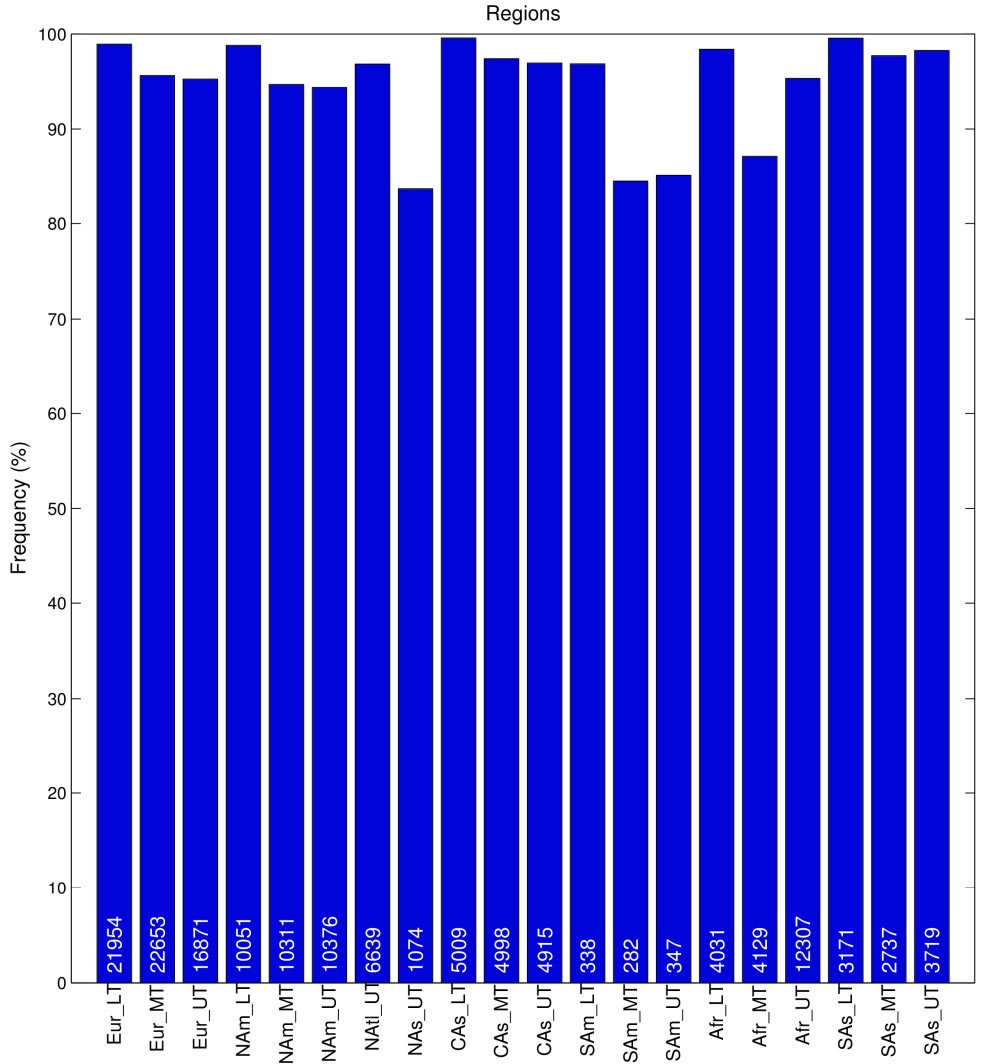

**Figure 9: Frequency of plume detection (a) in different regions / altitudes / seasons using the MACCity and GFAS v1.2 emission inventories during the 2003-2013 period. Biomass burning vertical injection uses APT methodology. Altitude levels stand for LT=0-2km, MT=2-8km and UT=8km-tropopause. The numbers of the plumes observed in each case are displayed in each box.**




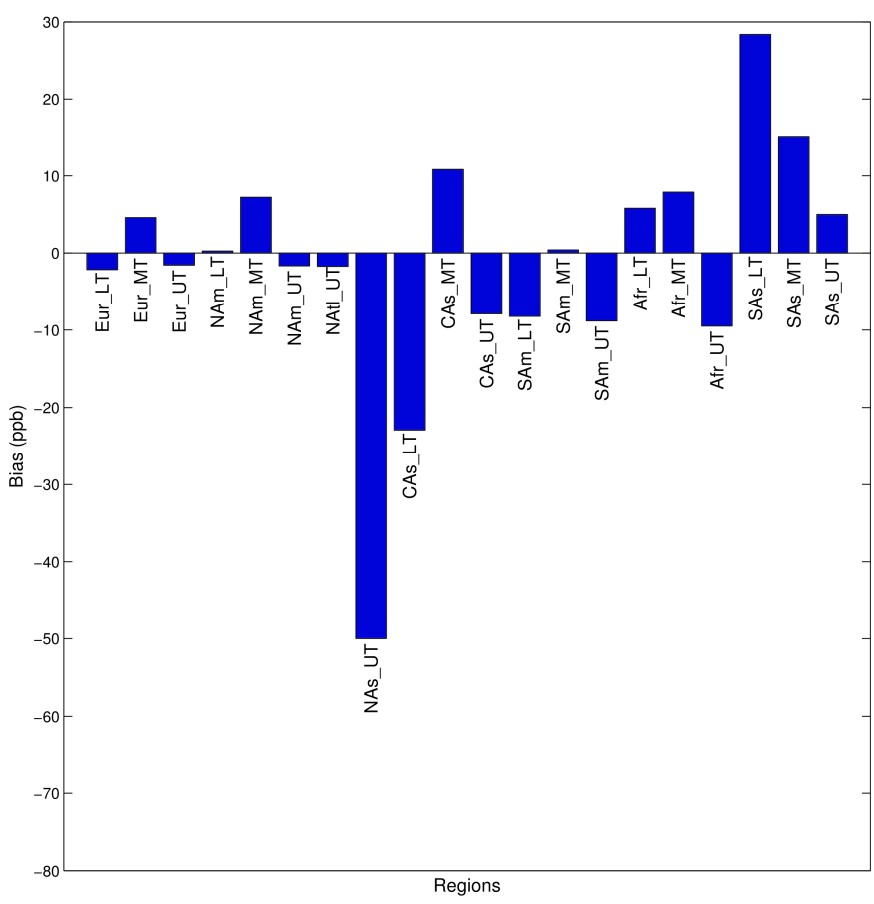

a)





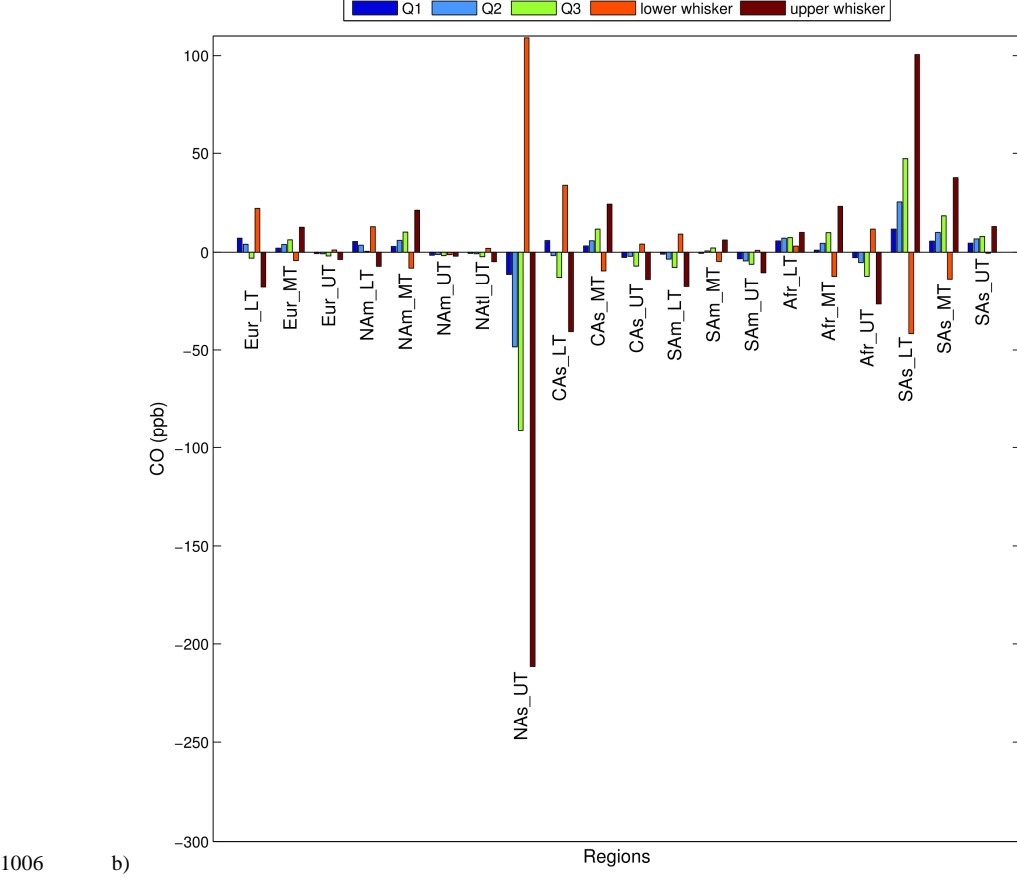

b)



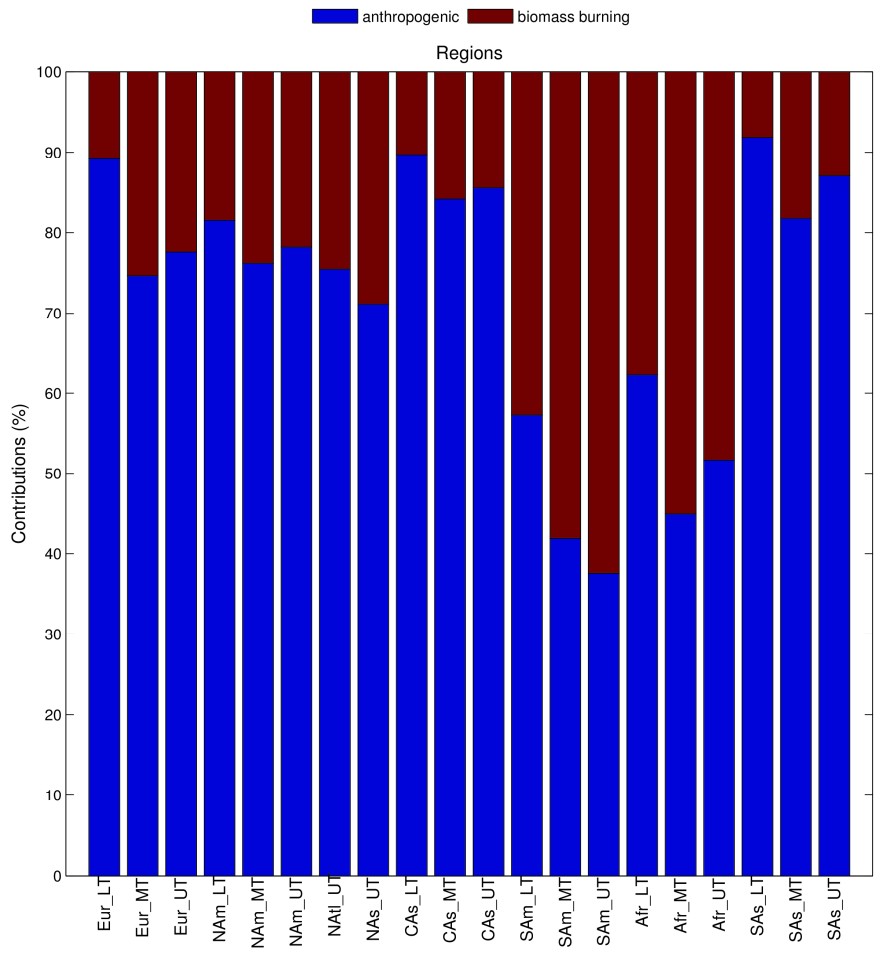

c)

**Figure 10: (a) Mean bias between the modeled and observed CO anomalies ; (b) Percentiles of the modeled CO anomalies bias with respect to observations; (c) Relative contribution from anthropogenic and biomass burning sources to the modeled CO. The three graphs are for the main sampled regions (Europe, North America, North Atlantic, North Asia, Central Asia, South America, Africa, South Asia) and in three layers (LT, MT, UT), using MACCity and GFASv1.2 for the 2003-2013 period. Biomass burning vertical injection uses APT methodology.**





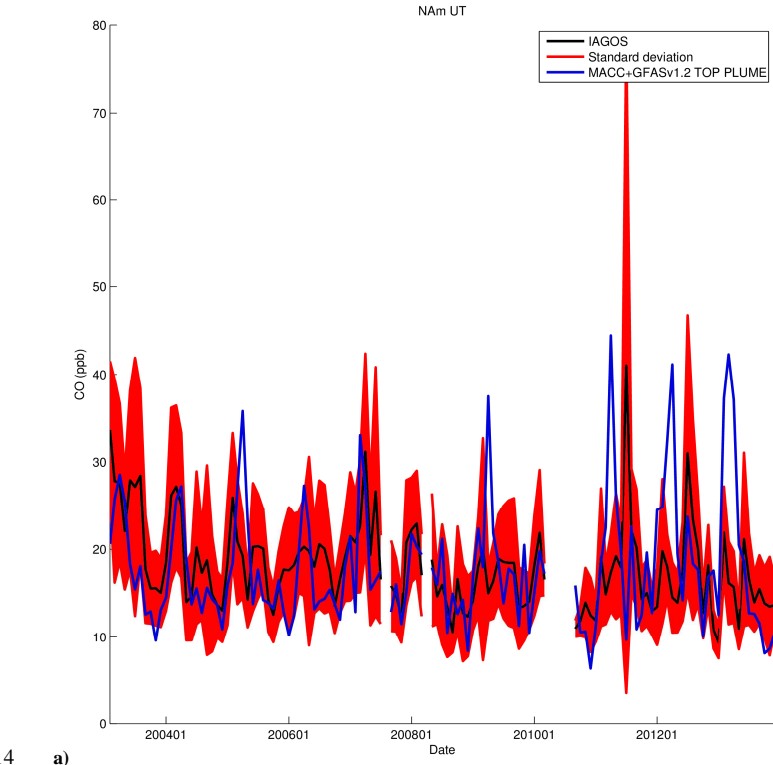

**a)**





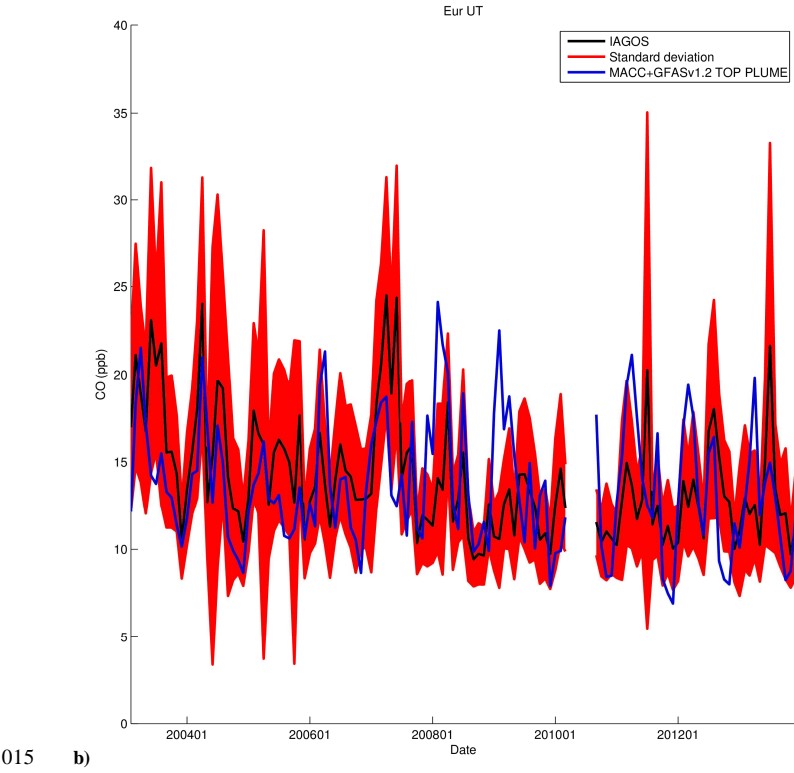

**b)**



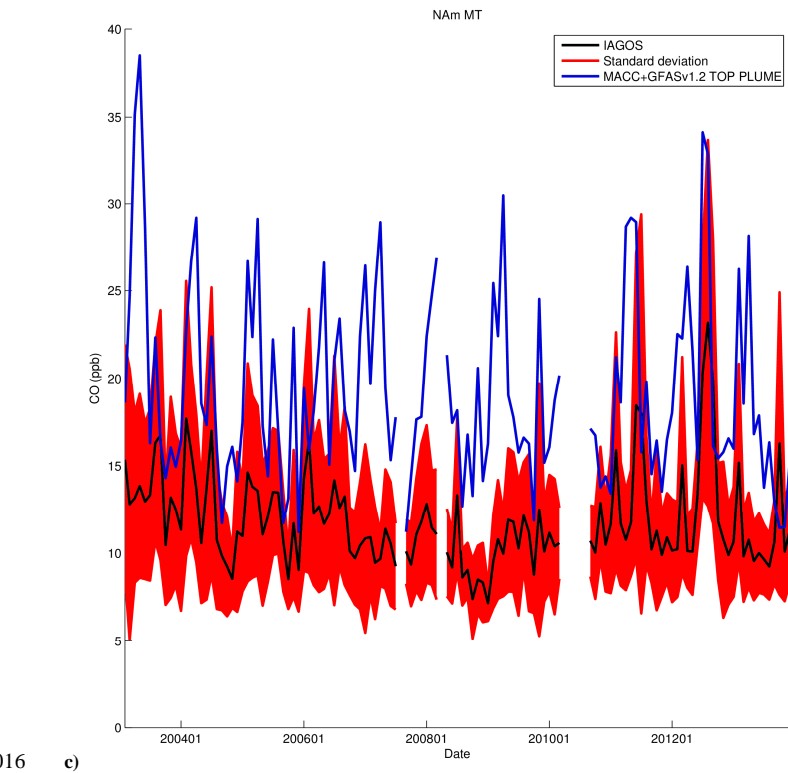

**c)**





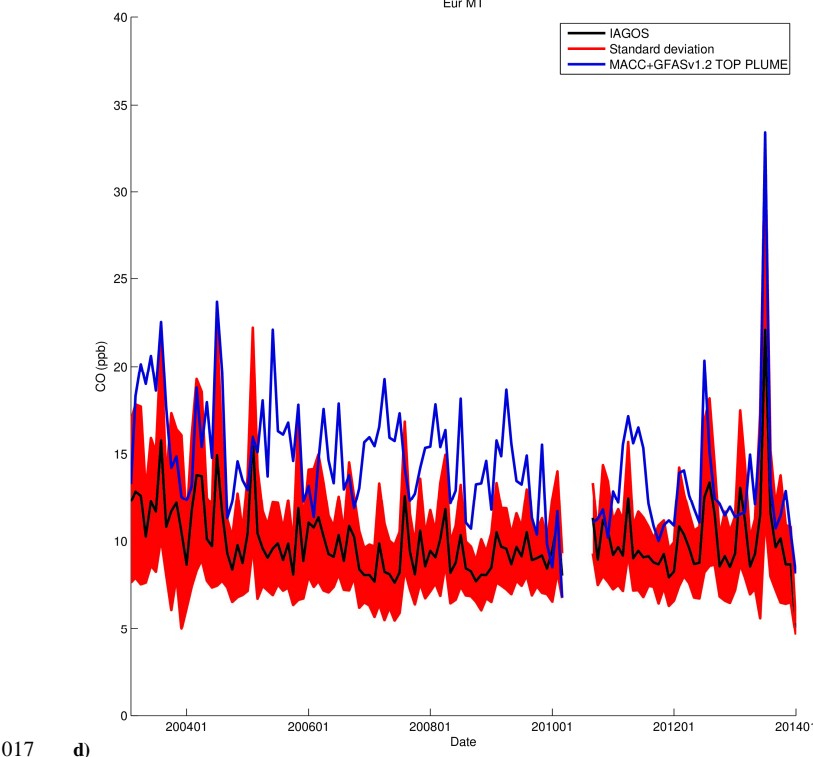

**d)**





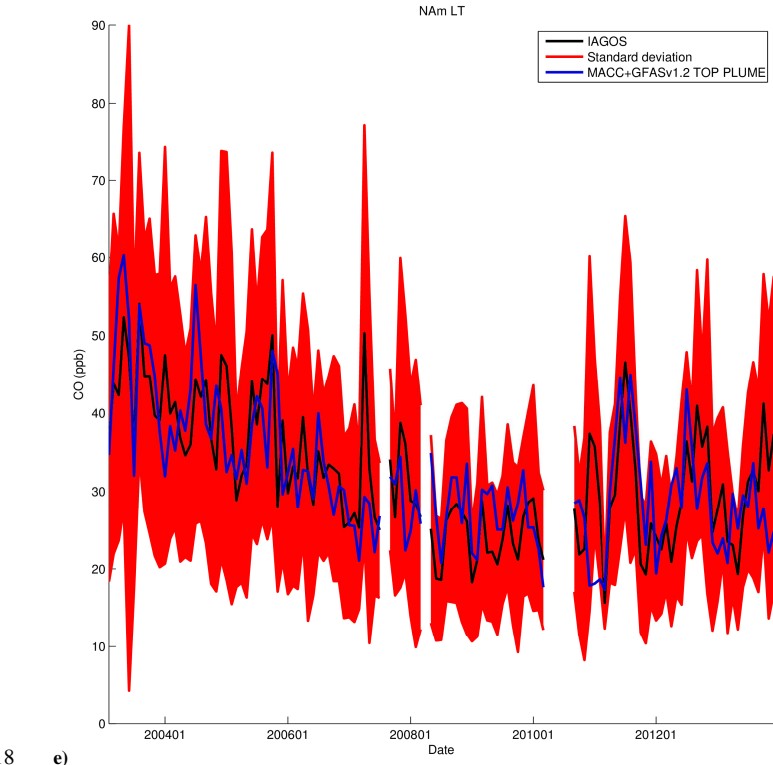

**e)**





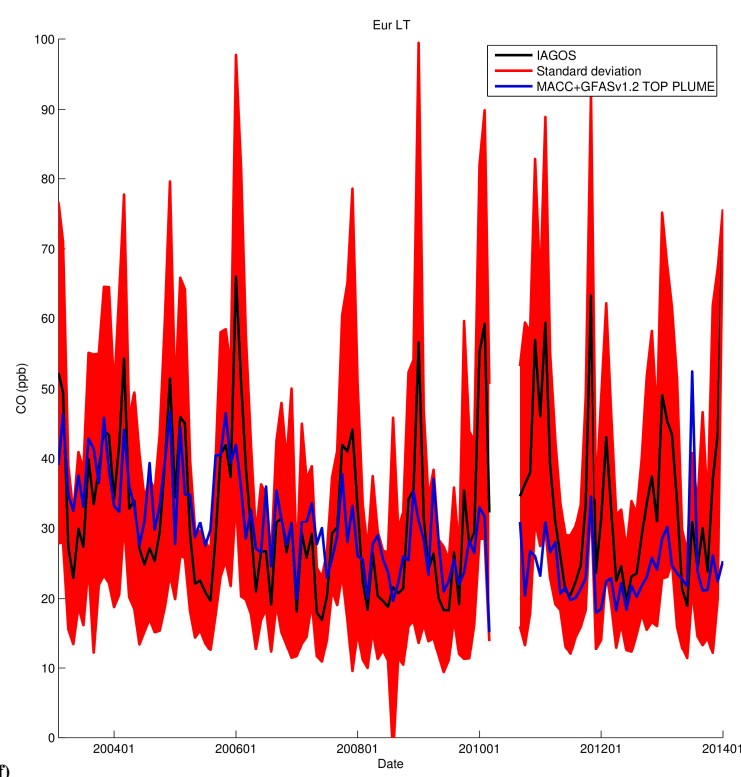

**f)**
**Figure 11: Times series (monthly means between 2003 and 2013) of the observed (black) and simulated (blue) plumes**
**of CO enhancements for the two most documented regions (North America and Europe) in the LT (e & f), MT (c & d)**
**and UT (a & b), using MACCity and GFASv1.2. Biomass burning vertical injection uses APT methodology. Red**
**shadow represents the standard deviation of the IAGOS observations**






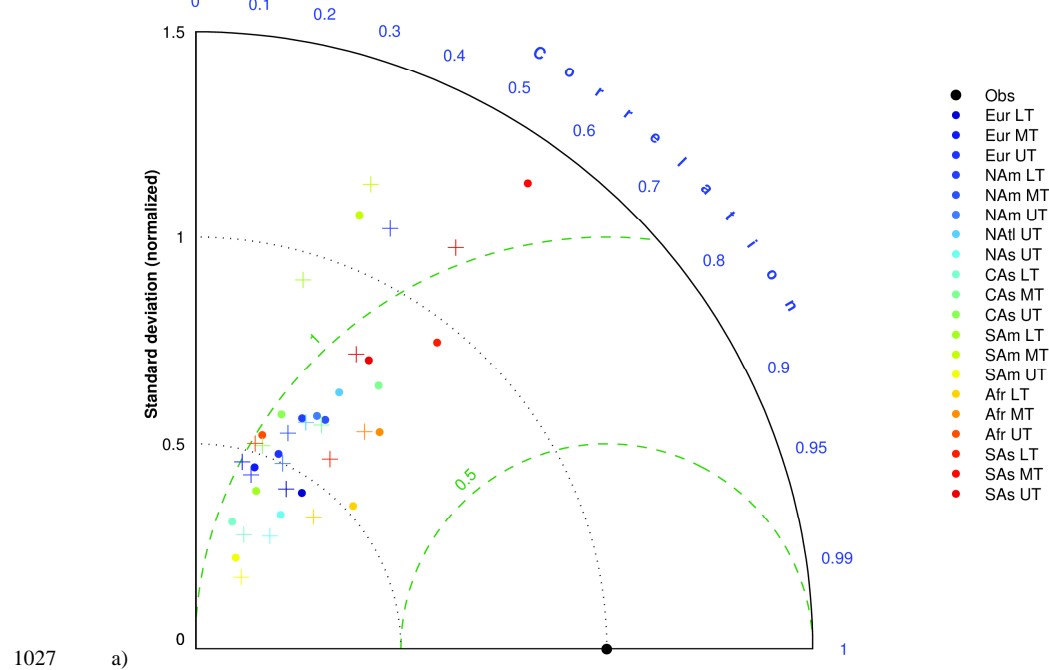

a)





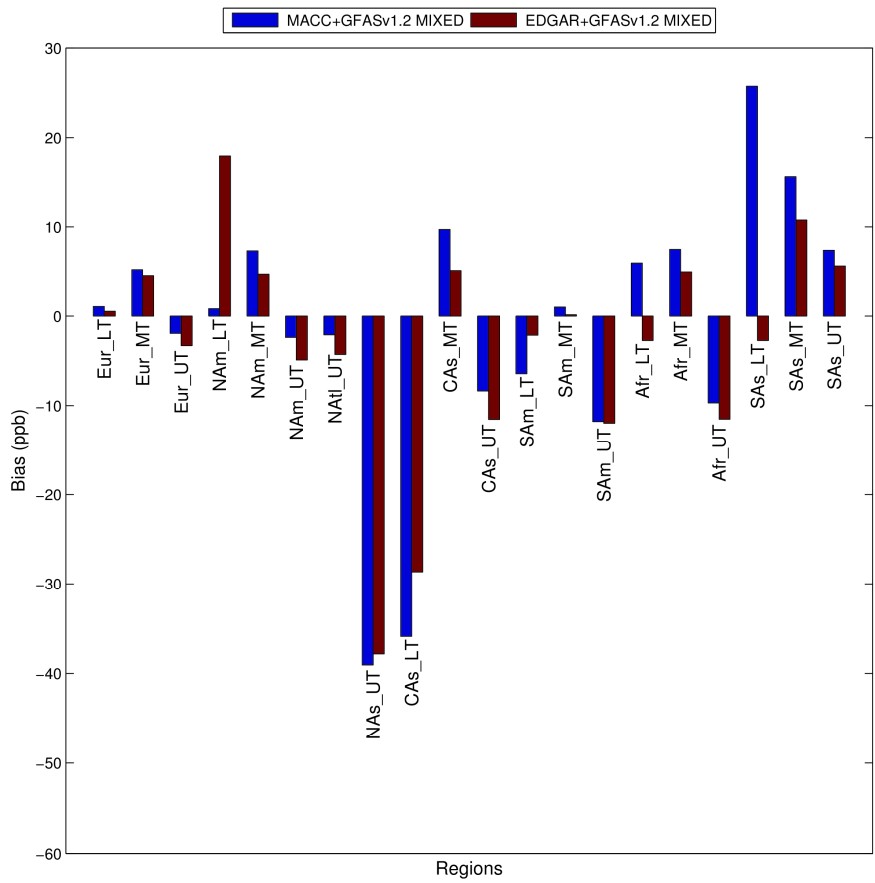

b)



**Figure 12: Comparison of the SOFT-IO anthropogenic emission influence between 2002 and 2008 (a) Taylor diagrams are obtained for the different regions and in the three vertical layers (LT, MT and UT) using MACCity (dots) and EDGARv4.2 (crosses) with GFAS (b) Mean biases between the modelled (blue for MACCity + GFAS; brown for EDGARv4.2 + GFAS) and observed CO anomalies. The MIXED methodology is used for fire vertical injection**





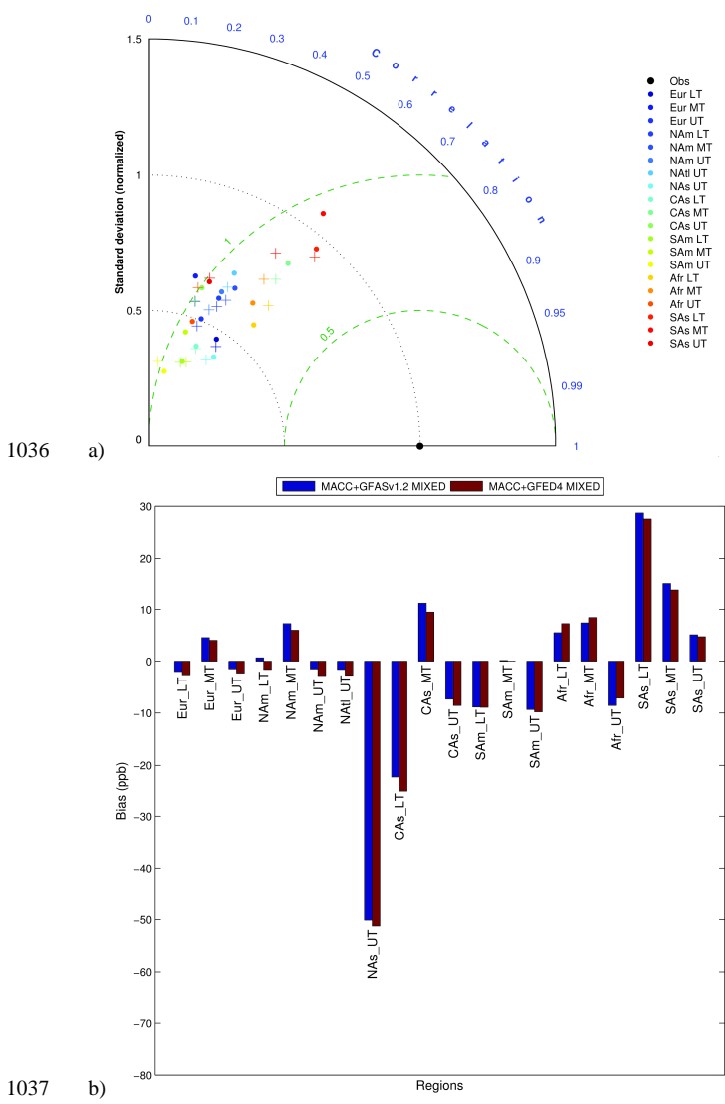





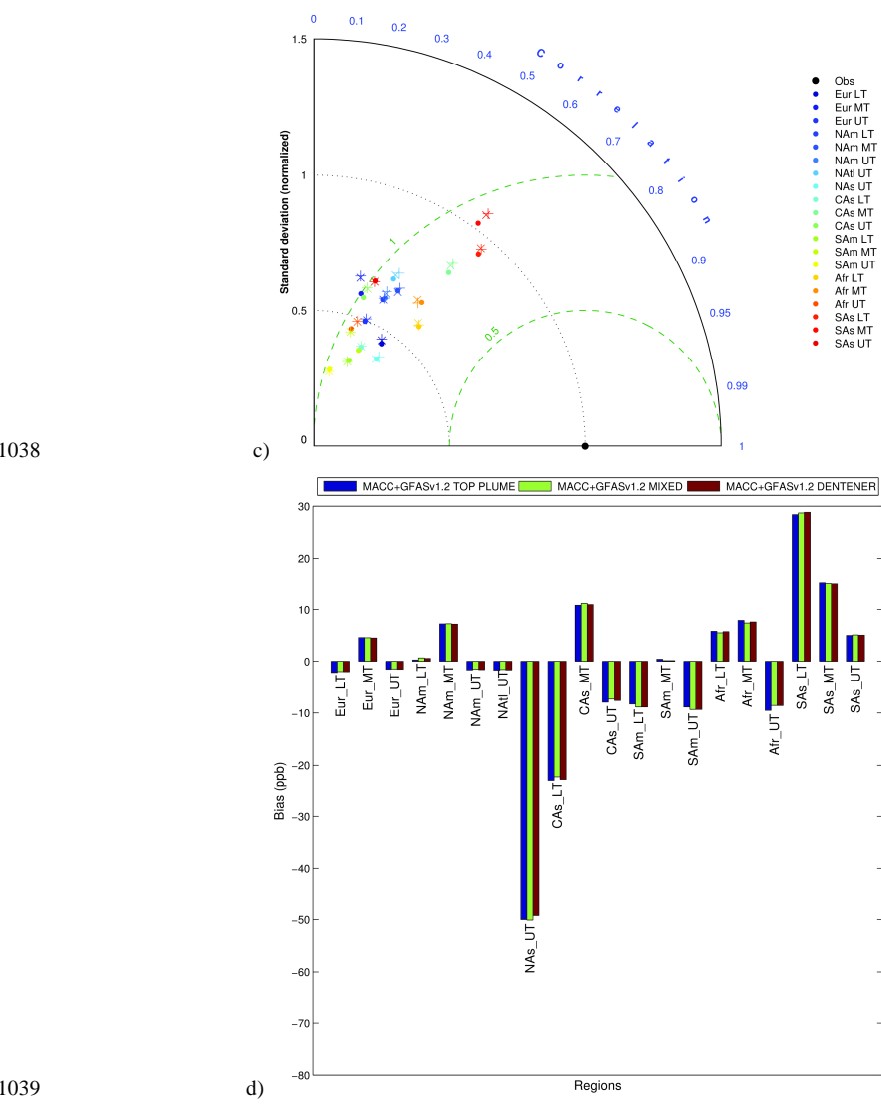

c)
d)

**Figure 13: Comparison of the SOFT-IO biomass burning emission influence between 2003 and 2013. Taylor diagrams are obtained for the different regions and in the three vertical layers (LT, MT and UT) using (a) GFASv1.2 (dots) and GFED4 (crosses) with MACCity and MIXED methodology for both GFASv1.2 and GFED4; (c) GFASv1.2 and MACCity with different vertical fire injections methodologies: MIXED (dots), APT (plus) and DENTENER (crosses). Mean biases between modeled and observed CO anomalies. Model is using (b) GFASv1.2 + MACCity (blue); GFED4 + MACCity (brown) and MIXED methodology for both GFASv1.2 and GFED4;   (d) GFASv1.2 + MACCity and different vertical fire injections methodologies: MIXED (blue); APT (green) and DENTENER (brown)**



