# Peer review of "Source attribution using FLEXPART and carbon monoxide emission inventories: SOFT-IO version 1.0"

_Atmospheric Chemistry and Physics, 2017_

## Referee Comment (RC1) · Anonymous Referee #1 · 29 Aug 2017

Review of "Source attribution using FLEXPART and 1 carbon monoxide emission inventories: SOFT-IO version 1.0" by Sauvage et al.

The paper documents the methodology and results from the use of FLEXPART on the IAGOS dataset, with the goal of providing potential users with source attribution. The paper is well-written and provide a good description of the methodology. The application portion of the paper is more limited, focusing on a few examples and broad measures. Overall, I find the paper worthy of publication after consideration of the following points.

Major point

While there is a wealth of information provided by all the parcels released along the flight track, the authors do not provide any information on the standard deviation (or any other statistical information) of the simulation perturbation. In particular, this seems to be of relevance to the discussion of Figure 11.

Minor points

- Line 162: It is not clear the vertical resolution is the most critical factor. Plenty of processes (as discussed in the paper) are not present in trajectories, or a choice of different parameters, could also be responsible for trajectory shortcomings.

- Line 208: Why the ICARTT dataset? There are plenty of regional dataset that might have been of higher relevance than this one. It would be good to justify this choice

- Line 220: it seems that the CO lifetime is not part of this equation. This would be a serious issue since 20-day trajectories are considered. If used, what is the CO lifetime?

- Line 228: it is also important to recognize the CO tends to be mostly released during smoldering and so might not be as prevalent in pyrocumuli.

- Line 286: it is not clear that it is always a straight linear decay with altitude.Âă How important is the definition of the background?

- Line 295: is there any assurance that the background from VP is consistent with UT where they connect? If not, is this an issue?

- Line 301: change "to consider" to "to be considered"

- Line 366: it would be nice to show PV along the same track

- Line 425: Figure needs an explanation of the color bar labels.

- Line 465: change "less good" to "worse"

- Line 471: I think it would be quite illuminating to present an additional figure (within the text or in the supplement) with percentages instead of concentrations.

- Line 488: this might look quite different with percentages!

- Line 497: this seems like a very narrow explanation.Ăă There are many things that could go wrong, not just pyro-cumulus.

- Line 502: I think "sense" is better than "information"

- Line 508: this seems like too many plots since very little discussion is attached to them

- Line 513: as mentioned in my major point above, the question is but what is the range of the variability from the different parcels?Ăă The only thing that this is showing is that the mean is within the observed standard deviation.

- Line 549: it is hard to get a sense of the change from the Taylor diagrams. If the authors want to keep them, it might be quite helpful to have arrows indicating the direction of the change.

- Line 555: this is actually incorrect. The anthropogenic emissions in MACCity originated from Lamarque et al. (ACP, 2010), except for the added seasonal cycle. Emissions were harmonized for year 2000 with the various scenarios (RCPs); therefore, any data post-2000 is actually the result of the scenario RCP8.5. The fact that they are fairly close is that they share many aspects (see paper above for more details).

———————————————————

---

## Referee Comment (RC2) · Anonymous Referee #2 · 16 Sep 2017

This paper by Sauvage et al., presents a system (SOFT-IO) based on the extensive use of FLEXPART dispersion model (coupled with different inventories of anthropogenic and fire emissions), created to analyse and attribute the variability of atmospheric composition observed along a huge number of observations by the IAGOS-MOZAIC programme. Even if, in this current configuration, the system is able to simulate only CO variability, it is valuable for the interpretation of this important long-term data base. From my understanding, the SOFT-IO outputs will be easily accessible to external users and thus they represent a potentially powerful tool for a number of applications. Since the system is based on a pre-computed data-set of air-mass transport simulation by FLEXPART model, it is possible to couple it with other emission inventories besides

those used in this work. As a personal comment, it would be really great if this system will be made available also for other observation systems (e.g. WMO/GAW stations).

Other than presenting SOFT-IO tool, the paper also provides an assessment of its performance in correcting reproducing the variability of observed CO due to anthropogenic and fire emissions over different World regions (where the IAGOS-MOZAIC programme is/was active) also discussing (by mean of case study analysis, and sensitivity studies) the dependency of SOFT-IO results as a function of different parameters (i.e. different input meteorological data-set, different emission inventories, different scheme for pyro-convection). By discussing the differences between SOFT-IO simulations and observations, the paper also provides information about the accuracy of different emission inventories or pyro-convection schema.

The paper is clear and very well written and I strongly recommend publication after that some points (most of them, minor) are considered. However, I have to stress (this is my only major concern) that the scientific significance of the SOFT-IO simulations are only limited discussed. As an instance, the authors provided very interesting long-term time series of CO over different regions of the World but without giving any comments or indications about differences among regions, about the existence/attribution of long-term trends (both in observations and simulations) , about seasonal variability or SOFT-IO agreement with other data-sets apart MOZAIC. In the same way, possible limitations/inaccuracy of the considered emission inventories (which have been pointed out by the authors) must be better addressed/discussed also in view of their extensive use in air-quality or climate studies.

Finally, I visited the IAGOS web site but I was not able to find SOFT-IO output. Probably, they are still not available to external users. . .

Minor/technical points 1) Figure 2: it seems that for boreal fires (with FRP > 10 Tjday) the injection fraction decrease with height along the first atmospheric layers (up to 2000 m). It is correct? This is the effect of atmospheric vertical mixing/stability?

2) In general the figure should be better arranged. I would recommend the authors to reshape the plots so that each full figure (often composed by several plates) can be showed in a single page. This would help the reader also in comparing the results of the sensitivity tests

3) Table 3: please provide some statistical indications to provide quantitative indication about the agreement for the two inventories (e.g. by providing average CO values for observations and simulations, mean bias, timing of the detected peak, std. dev..)

4) Pag 6. To me is not clear how the injection profile is defined...please clarify it.

5) Pag. 10. It's not clear why you claimed that only 2/3 of peaks are simulated by EDGAR. In my opinion, all the peaks are simulated by EGARD run indeed

6) Fig. 11, line 413. Thus the incorrect quantification of the bottom part of the peak by the ICARTT run can be attribute to not perfect transport/mixing by FLEXPART? Please comment, on that.

7) Pag 12, Figure 9: it can be interesting also to separate the plumes attributed to fires from these due to anthropogenic emissions .

8) Pag 13, line 493: I would say that for North Asia UT discrepancies varied from -100 to + 200 ppb and for South Asia LT from – 50 to +100 ppb.

9) Pag 14, line 516: the possible misrepresentation of anthropogenic emissions after 2009 is a point of great importance that deserve more discussion. The overestimation in the MT appeared to be more and more relevant over NAM than EU. Please comment.

10) Pag 15, line 559: I would not say that EDGAR performed better that MACC inventory for CAS_MT and NAS_UT: are these differences really significant?

---

## Author Comment (AC1) · 27 Oct 2017

Reviewer#1

The paper documents the methodology and results from the use of FLEXPART on the IAGOS dataset, with the goal of providing potential users with source attribution. The paper is well-written and provide a good description of the methodology. The application portion of the paper is more limited, focusing on a few examples and broad measures. Overall, I find the paper worthy of publication after consideration of the following points.

We would like to thank Reviewer#1 for her/his comments and suggestions that will improve our manuscript.
We clarified all the points raised by reviewer#1 and answered her/his different remarks in blue in this document.

Major point

While there is a wealth of information provided by all the parcels released along the flight track, the authors do not provide any information on the standard deviation (or any other statistical information) of the simulation perturbation. In particular, this seems to be of relevance to the discussion of Figure 11.

We provided statistical information in the submitted version through the percentiles information given in Fig 10b which are commented in Section 5.2.
In addition, as suggested by Rev#1, we have added in the revised version of the manuscript different statistical information.
SOFT-IO standard deviation has been added to Figure 11, as suggested by Rev#1, but also on Figs.#5 #6 #7 and #8 (see below for the modifications).
Additionally, we have also added standard deviation of the IAGOS vs SOFT-IO bias on Figure 10a, but not on Figs.12a and 13a for clarity reason.
The discussion related to the figures has been modified accordingly to take into account this new information on standard deviation in Section 5.2, as suggested by Rev.#1.

Minor points
- Line 162: It is not clear the vertical resolution is the most critical factor. Plenty of processes (as discussed in the paper) are not present in trajectories, or a choice of different parameters, could also be responsible for trajectory shortcomings.
We have modified line 162 in order to account Ref#1 remark:
" *Vertical resolution is one of the most critical factor for modeling such CO plumes with the best precision in terms of location and intensity (Eastham and Jacob, 2017)*"

- Line 208: Why the ICARTT dataset? There are plenty of regional dataset that might have been of higher relevance than this one. It would be good to justify this choice
Ref#1 is true that there are plenty of regional dataset that could have been tested. The goal of using regional dataset in the paper is to evaluate the incidence of one of them respect to global emission inventories, not to evaluate the incidence of all regional dataset. We have chosen ICARTT because of improved results demonstrated in the representation of boreal biomass burning fires in some specific cases (Elguindi et al., 2010; Turquety et al., 2016). Boreal fires

can be associated with pyro-convection, generally poorly represented in global emissions inventories. As IAGOS has a quasi global coverage, global emission inventories are the first choice in the methodology. However ICARTT comparison showed that regional inventories could be used to obtain better results on limited case studies on CO observations related to extreme events such as pyro-convection, and suggests that other regional emission inventories could be then included in the future in SOFT-IO for specific case studies CO pollution.
We have added the following sentence lines 206-209:
*"The aim is to test the ability of regional inventories in better representing simulated CO for specific case studies. The goal of using regional dataset in this paper is only to evaluate the incidence of one of them respect to global emission inventories, not to evaluate the incidence of all regional dataset. We have chosen ICARTT because of improved results demonstrated in the representation of boreal biomass burning fires in some specific cases (Turquety et al., 2016) as for example the one based on MOZAIC data by Elguindi et al., (2010). Global emission inventories are the first choice to interpret quasi global coverage of the CO IAGOS measurements. In the future we plan to include regional emission inventories for the study of specific events."*

- Line 220: it seems that the CO lifetime is not part of this equation. This would be a serious issue since 20-day trajectories are considered. If used, what is the CO lifetime?
CO is considered as chemically passive tracer in the equation. Concentrations will only vary considering dispersion and mixing associated with dynamical processes along 20 days.
The only significant chemical sink of CO in the troposphere is OH attack. As stated in lines 80-81, CO has lifetime of months in the troposphere (Logan et al., 1981; Mauzerall et al., 1998), higher than the 20-day of backtrajectories. Folkins et al. (JGR 2006) calculated CO lifetime against OH attacks (their Fig. 11) between 20-25 and 80 days within the troposphere, confirming that trajectories lower than 20-25 days should be used to avoid chemistry issues in CO lifetime.

- Line 228: it is also important to recognize the CO tends to be mostly released during smoldering and so might not be as prevalent in pyrocumuli.
The following sentence has been added line 228:
*"even if CO tends to be mostly released during smoldering"*

- Line 286: it is not clear that it is always a straight linear decay with altitude.Ǎa How important is the definition of the background?
We agree that there is not always a straight linear decay of CO with altitude. However, as for most of the IAGOS vertical profiles CO is enhanced in the boundary layer (related to surface emissions), the calculation of the background by using the slope calculated in the free troposphere was the most accurate way to define the background.
This definition of the background could be in the future improved by using "climatological" CO vertical profiles. It will be only possible to use this with sufficient CO measurements above the different IAGOS airports, and this was not possible for the present study over 10 years of CO measurements, except for few exceptions (Frankfurt for instance). Note that the definition of the background does not enter in the SOFT-IO methodology neither in the final CO ancillary data included in the IAGOS database. The background is defined in the present study to extract CO anomalies in order to statistically evaluate the differences with the contribution in CO computed by SOFT-IO. Finally the CO background definition has a negligible incidence in the CO anomalies definition, as we focus on the anomalies higher than the percentile 75 (see Eq. 4 and 5 lines 303-304)

- Line 295: is there any assurance that the background from VP is consistent with UT where they connect? If not, is this an issue?

Two different methodologies are used to estimate the background in UT and VP, as we still do not have enough data over all airports to apply climatological background for VP. Background is not used to provide ancillary data of CO in the IAGOS database and its definition is quite subjective (see for instance Parrish et al., 2012, doi:10.5194/acp-12-11485-2012 ). We estimate a background in the submitted paper to evaluate SOFT-IO simulations respect to CO anomalies events.

This is neither an issue for the provision of CO ancillary data calculated with SOFT-IO in the IAGOS database, nor for the estimation of CO anomalies as we focus on events higher than percentile 75, as explained just above.

- Line 301: change "to consider" to "to be considered"

Done

- Line 366: it would be nice to show PV along the same track

PV has been added in dark green along flight track on Figs.6a and 8a (see below)

- Line 425: Figure needs an explanation of the color bar labels.

Explanation of the color bar levels has been added  (see below)

- Line 465: change "less good" to "worse"

Change is done line 465

- Line 471: I think it would be quite illuminating to present an additional figure (within the text or in the supplement) with percentages instead of concentrations.

We have added additional figures of relative bias in supplement section (Figs S2a, S2b, S2c and S2d)

- Line 488: this might look quite different with percentages!

Figures with relative bias have been added in supplement (Fig S2a, S2b, S2c and S2d)

- Line 497: this seems like a very narrow explanation.Ă˘a There are many things that could go wrong, not just pyro-cumulus.

Rev#1 is true. We have added the following sentence line 497:

", or these emission inventories are under estimated for such specific events"

- Line 502: I think "sense" is better than "information"

Information has been replaced by sense

- Line 508: this seems like too many plots since very little discussion is attached to Them

Plots have been implemented over one page

- Line 513: as mentioned in my major point above, the question is but what is the range of the variability from the different parcels?Ă˘a The only thing that this is showing is that the mean is within the observed standard deviation.

As mentioned previously, we have added standard deviation into the figure and discussed it in Section 5.2. We clearly see that the standard deviation of the model is within the standard deviations of the observations in the LT and in the UT, but not in the MT.

- Line 549: it is hard to get a sense of the change from the Taylor diagrams. If the authors want to keep them, it might be quite helpful to have arrows indicating the direction of the change.
We have added connection lines to help the reader interpreting the direction of change in the Taylor diagrams (see below)

- Line 555: this is actually incorrect. The anthropogenic emissions in MACCity originated from Lamarque et al. (ACP, 2010), except for the added seasonal cycle. Emissions were harmonized for year 2000 with the various scenarios (RCPs); therefore, any data post-2000 is actually the result of the scenario RCP8.5. The fact that they are fairly close is that they share many aspects (see paper above for more details).

"

Rev#1 is true. We have updated information concerning MACCity in our manuscript in order to consider this remark. The following sentences have been added:

"*These results are not surprising as MACCity (Lamarque et al., 2010; Granier et al., 2011) is originated from various regional inventories (in addition to EDGAR), and expect to better represent...*"

[revised manuscript text omitted]

**Supplements**

[Figure]

**Figure S1: regions used to discriminate CO origin calculated with SOFT-IO, from**
**http://www.globalfiredata.org/data.html**

[Figure]

[Figure]

c)    d)

**Figure S2: Same as Figs. 10a, 12a, 13b and 13d (a, b, c, d respectively) but for relative bias (%)**

---

## Author Comment (AC2) · 27 Oct 2017

Reviewer#2

This paper by Sauvage et al., presents a system (SOFT-IO) based on the extensive use of FLEXPART dispersion model (coupled with different inventories of anthropogenic and fire emissions), created to analyse and attribute the variability of atmospheric composition observed along a huge number of observations by the IAGOS-MOZAIC programme. Even if, in this current configuration, the system is able to simulate only CO variability, it is valuable for the interpretation of this important long-term data base. From my understanding, the SOFT-IO outputs will be easily accessible to external users and thus they represent a potentially powerful tool for a number of applications. Since the system is based on a pre-computed data-set of air-mass transport simulation by FLEXPART model, it is possible to couple it with other emission inventories besides those used in this work. As a personal comment, it would be really great if this system will be made available also for other observation systems (e.g. WMO/GAW stations). Other than presenting SOFT-IO tool, the paper also provides an assessment of its performance in correcting reproducing the variability of observed CO due to anthropogenic and fire emissions over differentWorld regions (where the IAGOS-MOZAIC programme is/was active) also discussing (by mean of case study analysis, and sensitivity studies) the dependency of SOFT-IO results as a function of different parameters (i.e. different input meteorological data-set, different emission inventories, different scheme for pyroconvection).
By discussing the differences between SOFT-IO simulations and observations, the paper also provides information about the accuracy of different emission inventories or pyro-convection schema.
The paper is clear and very well written and I strongly recommend publication after that some points (most of them, minor) are considered.

We would like to thank Reviewer#2 for her/his comments and suggestions that will improve our manuscript. We clarified all the points raised by reviewer#2 and answered her/his different remarks or comments in blue in this document and in the revised manuscript.

However, I have to stress (this is my only major concern) that the scientific significance of the SOFT-IO simulations are only limited discussed. As an instance, the authors provided very interesting longterm time series of CO over different regions of the World but without giving any comments or indications about differences among regions, about the existence/attribution of long-term trends (both in observations and simulations) , about seasonal variability or SOFT-IO agreement with other data-sets apart MOZAIC.

Rev#2 is true that there is limited discussion of the scientific significance of SOFT-IO simulations. This choice is deliberate. Indeed as stated in the "Introduction" section (lines 86-90 of the submitted manuscript), the goal of the paper is to present and validate SOFT-IO as well as the CO ancillary products calculated with SOFT-IO, for the IAGOS database and the IAGOS users.
Ancillary products calculated with SOFT-IO will then be implemented in the IAGOS database, so that further scientific interpretations of the IAGOS data using SOFT-IO will follow in future papers realized by IAGOS database users.
For instance, long-term CO series have been first analyzed in a recent study of Cohen et al. (2017, https://doi.org/10.5194/acp-2017-778). Our study just aim to evaluate SOFT-IO in terms of long-term series reproducibility (Figure 11). The use of SOFT-IO to comment the existence/attribution, or to give indication about the differences in regional trends will be done in further studies, out of the scope of this paper. However, this is definitely our

objective, to go further for example regarding the CO trends analysis than the recent Cohen et al., 2017 work.

Moreover, as stated in the introduction, the main goal of SOFT-IO is to provide ancillary data that should help the IAGOS users interpreting the IAGOS database. SOFT-IO source code will be available as soon as the paper would be accepted, so that everybody could use it on other data-sets as suggested by Ref#2. We encourage external users to apply SOFT-IO to other dataset, such as ground based CO measurements, or for CO aircraft campaigns. However it is out of the scope of this study to evaluate the model to other dataset but IAGOS. Indeed IAGOS represents to our knowledge the densest in-situ measurements CO dataset, and it will be easier to apply SOFT-IO to other in situ CO datasets.

We modified the following sentence lines 86-90 for more clarity:
*"The goal is to provide the scientific community with added value products that will help them analyzing and interpreting the large number of IAGOS measurements. The methodology is focused on the development of a scientific tool (SOFT-IO version 1.0) based on FLEXPART particle dispersion model, that simulates the contributions of anthropogenic and biomass burning emissions for IAGOS CO measurements. This tool, which has the benefit to be adaptable to multiple emission inventories without re-running FLEXPART simulations, is described and then evaluated in the present study with the large data-sets of IAGOS CO measurements. SOFT-IO could be in the future easily adapted and used to analyze other datasets of trace gas measurements such as from ground based observations, sondes, aircraft campaigns or satellite observations."*

In the same way, possible limitations/inaccuracy of the considered emission inventories (which have been pointed out by the authors) must be better addressed/discussed also in view of their extensive use in air-quality or climate studies.

In the same way, we deliberately did not discuss the limitations and accuracy of the emission inventories. This is out of the scope of the paper. SOFT-IO could be in a future a useful tool to investigate emission inventories limitations or accuracy by the scientific community in charge of developing emission inventories, or investigating air quality or climate studies. The present paper only aims to present the SOFT-IO tool developed to help IAGOS users interpreting a large database such as the IAGOS one, to evaluate the tool against IAGOS data. Therefore, we provide these CO contributions to the IAGOS users as added-value products.

Finally, I visited the IAGOS web site but I was not able to find SOFT-IO output. Probably, they are still not available to external users: : :

Rev#2 is true, we believe that our validation paper should be accepted to make the code and the data available for external users.

Minor/technical points

1) Figure 2: it seems that for boreal fires (with FRP > 10 Tjday) the injection fraction decrease with height along the first atmospheric layers (up to 2000 m). It is correct? This is the effect of atmospheric vertical mixing/stability?

For boreal fires (> 10 and < 100TJ/day; > 100TJ/day), the injection fraction decreases with height higher than 3000m. Indeed this is the effect of atmospheric vertical mixing, as calculated by the PRMv2 model.

2) In general the figure should be better arranged. I would recommend the authors to reshape the plots so that each full figure (often composed by several plates) can be showed in a single page. This would help the reader also in comparing the results of the sensitivity tests

We have arranged the figures so that they are on a single page.

3) Table 3: please provide some statistical indications to provide quantitative indication about the agreement for the two inventories (e.g. by providing average CO values for observations and simulations, mean bias, timing of the detected peak, std. dev..)

We have added statistical information for Table 3 and Table 4 (see below)

4) Pag 6. To me is not clear how the injection profile is defined: : :please clarify it.

The injection profile is defined according to three methodologies, as explained page 7 lines 225-252, the DENTENER, the MIXED or the APT one.

In order to clarify, we add the following sentence lines 217-218:

*" and defined according to three different approaches (DENTENER, MIXED or APT) described in the next paragraph"*

5) Pag. 10. It's not clear why you claimed that only 2/3 of peaks are simulated by EDGAR. In my opinion, all the peaks are simulated by EGARD run indeed

All the peaks are simulated by EDGAR, but only 2/3 of the peaks intensity is reproduced using EDGAR.

We will rephrase lines 374-375

"*Only 2/3 of the observed enhancements are simulated using EDGARv4.2, except for plume 1 with better results*" with "*Using EDGARv4.2, only 2/3 of the observed CO enhancements intensity is reproduced, except for plume #1 with better intensity results*"

6) Fig. 11, line 413. Thus the incorrect quantification of the bottom part of the peak by the ICARTT run can be attribute to not perfect transport/mixing by FLEXPART? Please comment, on that.

It seems that Rev#2 refers here to Fig.7a. In this case it is hard to explain why the bottom part of the peak is not represented as well as the upper part, either by ICARTT, GFED or GFAS. It could indeed be related to transport processes in FLEXPART, but also in the ECMWF analyses or in the vertical profiles injection.

7) Pag 12, Figure 9: it can be interesting also to separate the plumes attributed to fires from these due to anthropogenic emissions .

Ideally this could be interesting. But this is not possible to realize. Indeed all the plumes are influenced both by biomass burning and anthropogenic emissions, as we can see on the case studies displayed on Figures 5 to 8. In order to do that we should define subjective criteria to attribute a plume to either biomass burning or anthropogenic emissions. This is out of the scope of this study.

8) Pag 13, line 493: I would say that for North Asia UT discrepancies varied from -100 to + 200 ppb and for South Asia LT from – 50 to +100 ppb.

We have modified line 493 with Rev#2 suggestion:

"*North Asia UT discrepancies varies from -100 ppb to +200 ppb and from -50 ppb to +100 ppb for South Asia LT.* "

9) Pag 14, line 516: the possible misrepresentation of anthropogenic emissions after 2009 is a point of great importance that deserve more discussion. The overestimation

in the MT appeared to be more and more relevant over NAM than EU. Please comment.
Rev#2 is true. As stated in Stein et al., 2004, the largest near-surface CO bias are found over Europe in January.
We have added the following lines 519:
*"This suggests misrepresentation of anthropogenic emissions in Europe after the year 2009. Indeed Stein et al., (2014) suggested the lower near-surface CO bias was found in Europe in relation with possible under estimation of traffic emissions in the inventories."*

It is also true that the overestimation in the MT appears higher over NAM rather than over EU. This could be related to two causes:
- Less measurements in the MT over NAM than over EU
- Greater proximity of the NAM MT to summer sources, such as boreal fires, that could explain the higher overestimation particularly in this season.

We add the following lines 519-529
*"In the middle troposphere (2-8 km), the CO plumes are systematically overestimated by SOFT-IO by 50% to 100% compared to the observations, with larger standard deviation and higher overestimation over NAm. This might be related to different reasons:*

- *the chosen methodology of the CO plume enhancements detection for those altitudes (described in Sect. 3.4), which may lead to a large number of plumes with small CO enhancements, which are difficult to simulate. This could be due to the difficulty in defining a realistic CO background in the middle troposphere.*

- *the source-receptor transport which may be more difficult to simulate between 2-8 km than in the LT where receptors are close to sources; or than in the UT where most of the plumes are related to convection detrainment better represented in the models than MT detrainment which might be less intense.*

- *The frequency of the IAGOS observations which is lower in the LT and in the MT than in the UT.*

- *Higher overestimation over NAm MT than Eur MT could be first related to lower frequency of measurements in the NAm. Moreover overestimation is greater during summer when NAm MT is closer to summer sources such as boreal fires, while Eur MT is related to CO air masses more diluted with background air during transatlantic transport."*

10) Pag 15, line 559: I would not say that EDGAR performed better that MACC inventory for CAS_MT and NAS_UT: are these differences really significant?
Indeed results are better using EDGAR for specific regions. Ref#2 is right. Differences are not statistically significant for NAs_UT, but they are for CAs_MT (almost 50% difference between the two simulations with the two inventories).
We rephrase line 559 with the following *"Regionally, however, results with EDGARv4.2 can be better by almost 50%, such as over South Asia LT and MT, Central Asia LT and MT"*

**Tables modifications requested by Rev#2:**

| Flight | IAGOS anomaly | IAGOS std | MACCity anomaly | MACCity std | EDGAR anomaly | EDGAR std | Anomaly altitude |
|---|---|---|---|---|---|---|---|
| 10 March 2002 Frankfurt – Denver | 16.8 | 8.7 | 20.2 | 6.9 | 12.8 | 5.1 | UT |
| 27 November 2002 Dallas – Frankfurt | 28.0 | 8.6 | 20.0 | 8.0 | 16.4 | 7.4 | UT |
| 19 July 2005 München - Hong Kong | 130.1 | 97.8 | 45.8 | 9.7 | 34.6 | 7.7 | PBL |
| 22 October 2005 München - Hong Kong | 157.9 | 105.1 | 170.7 | 109.8 | 103.9 | 62.0 | PBL |

**Table 3. Summary of the averaged observed and simulated anomaly and corresponding averaged standard deviation (std) (in ppb) determined for representing anthropogenic emissions for different case studies (using GFAS v1.2 for biomass burning emissions). Altitude of the anomaly is indicated: boundary layer (PBL); middle troposphere (MT); upper troposphere (UT)**

| Flight | IAGOS anomaly | IAGOS std | GFAS v1.2 anomaly | GFAS v1.2 std | GFED4 anomaly | GFED4 std | ICARTT anomaly | ICARTT std | Anomaly altitude |
|---|---|---|---|---|---|---|---|---|---|
| 29 June 2004 Caracas - Frankfurt | 32.6 | 33.2 | 44.4 | 2.4 | 43.0 | 2.3 | 43.6 | 2.4 | PBL |
| 30 June 2004 Frankfurt - Washington | 52.5 | 34.0 | 36.6 | 9.1 | 25.4 | 6.6 | 23.5 | 5.9 | MT |
| 22 July 2004 Frankfurt - Atlanta | 87.0 | 35.0 | 42.8 | 17.6 | 45.8 | 18.9 | 39.7 | 15.7 | MT |
| 22 July 2004 Douala - Paris | 117.1 | 24.2 | 43.5 | 20.0 | 55.0 | 27.2 | 72.4 | 42.3 | MT |
| 23 July 2004 Frankfurt - Atlanta | 78.9 | 45.4 | 34.7 | 22.4 | 45.3 | 32.8 | 46.0 | 35.9 | MT |
| 30 July 2008 Windhoek - Frankfurt | 72.9 | 41.9 | 33.0 | 19.2 | 42.8 | 26.0 | N/A | N/A | UT |
| 31 July 2008 Frankfurt - Windhoek | 38.3 | 32.0 | 28.1 | 10.8 | 34.0 | 12.8 | N/A | N/A | UT |

**Table 4. Summary of the averaged observed and simulated anomaly and corresponding averaged standard deviation (std) (in ppb) determined for representing biomass burning emissions for different case studies (using MACCity for anthropogenic emissions). Altitude of the anomaly is indicated: boundary layer (PBL); middle troposphere (MT); upper troposphere (UT).** Note that the ICARTT inventory is only available for summer 2004.